# Gradients with Respect to Semantics Preserving Embeddings Tell the Uncertainty of Large Language Models

## Abstract

Uncertainty quantification (UQ) is an important technique for ensuring the trustworthiness of LLMs, given their tendency to hallucinate. Existing state-of-the-art UQ approaches for free-form generation rely heavily on sampling, which incurs high computational cost and variance. In this work, we propose the first gradient-based UQ method for free-form generation, SemGrad, which is sampling-free and computationally efficient. Unlike previous gradient-based methods developed for classification tasks, we propose to operate in semantic space rather than parameter space. Our method builds on the key intuition that a confident LLM should maintain stable output distributions under semantically equivalent input perturbations. We interpret the stability as the gradients in semantic space and introduce a Semantic Preservation Score (SPS) to identify embeddings that best capture semantics, with respect to which gradients are computed. We further propose HybridGrad, which combines the strengths of SemGrad and parameter gradients. Experiments demonstrate that both of our methods provide efficient and effective uncertainty estimates, achieving superior performance than state-of-the-art methods, particularly in settings with multiple valid responses.

## 1 Introduction

With the widespread deployment of Large Language Models (LLMs) across various domains, including education, healthcare, and finance (Naveed et al., 2023; Zhao et al., 2023; Chiarello et al., 2024; Raza et al., 2025), the reliability of their responses has become a pressing concern. Despite their impressive generative abilities, LLMs remain prone to hallucinating untruthful contents which undermine credibility in real-world applications (Zhang et al., 2023; Huang et al., 2024). Uncertainty Quantification (UQ) has emerged as a promising approach to mitigate these risks by providing not only what models predict, but also how confident they are in those predictions (Baan et al., 2023; Shorinwa et al., 2024).

Although Uncertainty Quantification has been widely explored and proven effective in classification tasks (Gawlikowski et al., 2023), extending it to LLM-based free-form generation presents unique challenges. Unlike standard classification, where the label space is fixed and relatively constrained, LLMs operate in a sequential classification framework with an extremely large vocabulary at each step, resulting in a combinatorially vast output space. Moreover, the inherent nature of natural language allows multiple valid responses for a single input, introducing a substantially higher degree of aleatoric uncertainty—defined as the inherent, irreducible randomness within the data (Hüllermeier & Waegeman, 2021)—than is typically observed in single-step classification tasks (Baan et al., 2023). State-of-the-art UQ methods for free-form generation primarily rely on sampling-based approaches that capture semantic variation within the output space (Kuhn et al., 2023; Chen et al., 2024; Duan et al., 2024; Qiu & Miikkulainen, 2024). Although these approaches generally outperform self-verbalized methods or simple deterministic logit-based methods (Shorinwa et al., 2024; Kuhn et al., 2023), they require a substantial number of samples to approximate the vast output space, resulting in high variance and computational cost, especially given the scale of current LLMs.

Unlike sampling-based methods, gradient-based methods directly exploit gradients of the log probability of generated outputs, which can be collected in parallel with the generation process, en-

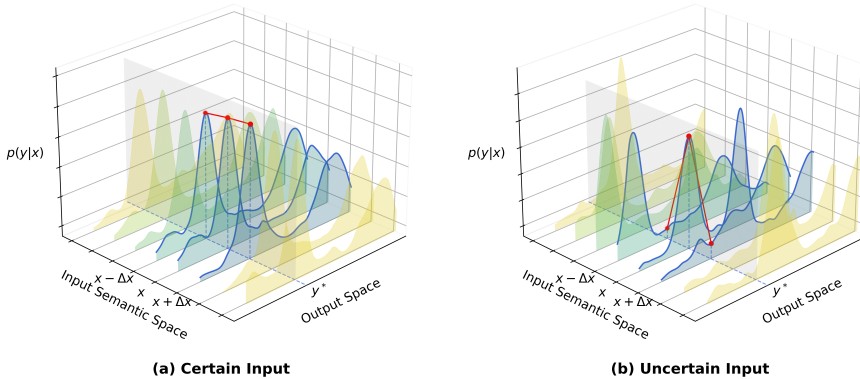

**(a) Certain Input**          **(b) Uncertain Input**

Figure 1: Illustration of output distribution shift under small input semantic perturbations and the semantic gradients. $\boldsymbol{x}$ represents the original input, and $\boldsymbol{x} + \Delta\boldsymbol{x}$ denotes a perturbed input with a small semantic change on $\boldsymbol{x}$ in the semantic space. $\boldsymbol{y}^*$ denotes the response generated from $p(\boldsymbol{y}|\boldsymbol{x})$. For an input that the model is certain about, a small semantic perturbation should not significantly alter the output distribution, as shown in (a), i.e., $p(\boldsymbol{y}^*|\boldsymbol{x})$ is insensitive to small semantic perturbation. The sensitivity can be captured by the magnitude of the slope of the red line, corresponding to the gradient in semantic space when $\Delta\boldsymbol{x} \to 0$. In contrast, the gradient will be high for the uncertain input, as shown in (b).

abling sampling-free and efficient estimation. However, previous work (Lee & AlRegib, 2020; Igoe et al., 2022; Wang & Ji, 2024) on gradient-based UQ was developed for classification tasks with the assumption that each input has a single ground-truth label (i.e., zero aleatoric uncertainty). This assumption breaks down in free-form generation, where valid responses are not always unique. Moreover, the sequential nature of generation complicates UQ, since individual tokens contribute unequally to meaning (Duan et al., 2024), with some carrying high semantic weight while others are negligible. Therefore, a gradient-based method specifically designed for free-form generation is needed.

In this work, we introduce, to our knowledge, the first gradient-based approach to uncertainty quantification for free-form generation in LLMs. Unlike prior work that measures gradients in parameter space, we consider gradients in semantic space. The underlying intuition is straightforward: if a well-trained LLM is confident in its response to a query, its output distribution should remain stable when the query is perturbed with semantically equivalent variants (Figure 1 (a)). This mirrors human behavior: when confident, people tend to provide consistent answers, whereas uncertainty often leads to variability in responses. Building on this assumption, we quantify uncertainty by measuring how sensitively the output probabilities change under small semantic perturbations. Technically, this sensitivity can be described by the gradient of the output probability with respect to the semantic preserving embeddings (slope of the red line in Figure 1). We call this method Semantic Gradients (**SemGrad**). Notably, our method does not rely on any assumption about the form of the ground-truth distribution and thus remains valid even under high aleatoric uncertainty.

To identify the embeddings that best preserve input semantic information, we introduce the Semantic Preservation Score (SPS), which measures the alignment difference of each hidden state between semantic-equivalent paraphrases and semantically different ones, and identify the semantic preserving embeddings as the embeddings with high SPS. Meanwhile, to mitigate the issue of linguistic redundancy, we further propose a simple yet effective method that re-weights the output probabilities prior to gradient computation. While parameter gradients can be unreliable under high aleatoric uncertainty, they remain competitive in single–ground-truth settings. To leverage the advantages of both approaches and improve generalization, we propose a hybrid metric named **HybridGrad**. Our experiments on several QA benchmarks demonstrate that both SemGrad and HybridGrad provide efficient and effective uncertainty estimates, achieving superior performance than state-of-the-art methods, particularly in cases where multiple responses are correct. Meanwhile, our experiments reveal a strong positive correlation between the capability of hidden states to preserve semantics and

the UQ performance of SemGrad when gradients are computed with respect to them. This finding supports our claim that the method indeed operates in semantic space, consistent with our assumption that the output distribution of an LLM should remain stable under small semantic perturbations of the input, but not under arbitrary random perturbations.

## 2 PRELIMINARIES

In this section, we provide a brief overview of the architecture of prevailing large language models, introduce the notations and key concepts used throughout the paper, and finally review gradient-based uncertainty quantification methods developed for classification tasks.

Current LLMs generally follow a causal autoregressive paradigm. Given an input text $\boldsymbol{x} = \{x_1, x_2, ..., x_I\}$, where each $x_i$ represents an input token, a causal language model factorizes the probability of generating a response $\boldsymbol{y} = \{y_1, y_2, ..., y_T\}$ into conditional distributions,

$$p(\boldsymbol{y}|\boldsymbol{x}, \boldsymbol{\theta}) = \prod_{t=1}^{T} p(y_t|y_{<t}, \boldsymbol{x}, \boldsymbol{\theta})$$

and generates tokens in a left-to-right manner. When generating the token $y_{t+1}$, each input token $x_i$ and previously generated token $y_{\leq t}$ are mapped to a sequence of embeddings, one per token, yielding the initial hidden states $\boldsymbol{h}^{(0)}$. The initial hidden states are then processed by a stack of $L$ transformer blocks. At each layer $l$, the hidden states are updated through a residual connection around a self-attention transformation followed by another residual connection around a feed-forward transformation,

$$\boldsymbol{h}_j^{(l)} = \boldsymbol{h}_j^{(l-1)} + \text{Attn}(\boldsymbol{h}_{\leq j}^{(l-1)}) + \text{FFN}(\boldsymbol{h}_j^{(l-1)} + \text{Attn}(\boldsymbol{h}_{\leq j}^{(l-1)})) , \ j \leq t$$

where attention operates on previous tokens by a causal mask. We omit normalization terms for brevity, as they vary across structures and are not central to our work. After traversing all $L$ layers, the final hidden state $\boldsymbol{h}_t^{(L)}$ is projected into vocabulary space by the LM head weight $\boldsymbol{W}_{\text{head}}$ to produce logits, and then get the output distribution through softmax,

$$\boldsymbol{z}_t = \boldsymbol{h}_t^{(L)} \boldsymbol{W}_{\text{head}}^T ; \quad p(y_{t+1}|y_{\leq t}, \boldsymbol{x}, \boldsymbol{\theta}) = \text{Softmax}(\boldsymbol{z}_t)$$

We use $\boldsymbol{\theta}$ to denote all parameters of an LLM, including the weight matrices (and bias if any) in each attention and FFN layer, as well as the token embedding matrix $\boldsymbol{E}$, LM head matrix $\boldsymbol{W}_{\text{head}}$ and parameters in normalization.

Training an LM aims to approximate the ground truth human language distribution $p^*$. Accordingly, we minimize the expected negative log-likelihood of sequences under $p^*$,

$$\min_{\boldsymbol{\theta}} \mathbb{E}_{\boldsymbol{x} \sim p^*(\boldsymbol{x})}[\mathbb{E}_{\boldsymbol{y} \sim p^*(\boldsymbol{y}|\boldsymbol{x})}[-\log p(\boldsymbol{y}|\boldsymbol{x}, \boldsymbol{\theta})]]$$

For a specific input $x$, if $\boldsymbol{\theta}^*$ is optimal, it should minimize $\mathbb{E}_{\boldsymbol{y} \sim p^*(\boldsymbol{y}|\boldsymbol{x})}[-\log p(\boldsymbol{y}|\boldsymbol{x}, \boldsymbol{\theta})]$, and therefore its gradient with respect to $\boldsymbol{\theta}$ is vanished at optimal $\boldsymbol{\theta}^*$,

$$\nabla_{\boldsymbol{\theta}} \mathbb{E}_{\boldsymbol{y} \sim p^*(\boldsymbol{y}|\boldsymbol{x})}[-\log p(\boldsymbol{y}|\boldsymbol{x}, \boldsymbol{\theta})]\big|_{\boldsymbol{\theta}=\boldsymbol{\theta}^*} = 0 \tag{1}$$

In a classification task, if we assume that there exists only a single ground-truth label $y^*$ (i.e., zero aleatoric uncertainty), then the ground-truth distribution $p^*$ degenerates to a Dirac delta distribution. In this case, the expectation $\mathbb{E}_{\boldsymbol{y} \sim p^*(\boldsymbol{y}|\boldsymbol{x})}[-\log p(\boldsymbol{y}|\boldsymbol{x}, \boldsymbol{\theta})]$ collapse to $-\log p(y^*|\boldsymbol{x}, \boldsymbol{\theta})$, and the gradient equation 1 reduces to

$$\nabla_{\boldsymbol{\theta}} \log p(y^*|\boldsymbol{x}, \boldsymbol{\theta})\big|_{\boldsymbol{\theta}=\boldsymbol{\theta}^*} = 0 \tag{2}$$

This observation motivates the use of parameter gradient norm $||\nabla_{\boldsymbol{\theta}} \log p(\boldsymbol{y}|\boldsymbol{x}, \boldsymbol{\theta}_{\mathcal{M}})||$ as a proxy for UQ of a model $\mathcal{M}$ on classification tasks (Igoe et al., 2022; Wang & Ji, 2024). A small value indicates that the model is well trained on the given data point and confident in its prediction, whereas a large value suggests the opposite, i.e., higher uncertainty.

However, this reasoning does not extend to the ground-truth distribution of natural language, $p^*(\boldsymbol{y}|\boldsymbol{x})$, which usually exhibits high aleatoric uncertainty due to the existence of multiple valid responses. In this setting, equation 2 is not necessarily satisfied even at an optimum because $p^*(\boldsymbol{y}|\boldsymbol{x})$ is no longer a Dirac delta. As a result, the parameter gradient norm can be misleading, as large values may reflect the aleatoric uncertainty of the task rather than genuine model uncertainty. Meanwhile, approximating the expectation of equation 1 is computationally intractable and inefficient for modern LLMs, due to their extremely large output space and parameter size.

# 3 SEMANTIC GRADIENTS

To overcome the limitation of the parameter gradient, we propose to evaluate gradients with respect to the semantic space, inspired by an intrinsic nature of human language: stable input semantics should yield stable output semantics.

## 3.1 WHY GRADIENTS ON SEMANTICS?

We start from a simple assumption about human language: **no matter how syntactic form may vary, as long as the underlying context meaning (semantics) is preserved, the responses should remain stable**. Accordingly, for the ground-truth distribution of human language $p^*(\boldsymbol{y}|\boldsymbol{x})$, we assume that semantically equivalent inputs $\boldsymbol{x}$ and $\boldsymbol{x}'$ yield a similar output distribution, i.e.,

$$p^*(\boldsymbol{y}|\boldsymbol{x}) \approx p^*(\boldsymbol{y}|\boldsymbol{x}')$$

In other words, the true distribution should be insensitive to small perturbations in the semantic space. This mirrors human behavior — when confident, people tend to provide consistent answers, whereas uncertainty often leads to variability in responses.

Now consider an LLM $p(\boldsymbol{y}|\boldsymbol{x}, \boldsymbol{\theta})$, which generates a specific output $\hat{\boldsymbol{y}}$ given input $\boldsymbol{x}$. Suppose we can identify a semantic-preserving embedding $\boldsymbol{h}_E(\boldsymbol{x})$, such that semantically equivalent variants of $\boldsymbol{x}$ are mapped to nearby vectors, while semantically different inputs are mapped to distant ones. A perturbation on $\boldsymbol{h}_E(\boldsymbol{x})$, i.e., $\boldsymbol{h}_E(\boldsymbol{x}) + \Delta\boldsymbol{h}_E$, can then be regarded as a semantic variation of $\boldsymbol{x}$. As long as $\Delta\boldsymbol{h}_E$ is sufficiently small, the perturbation should preserve semantics. If the LM is well-trained on $\boldsymbol{x}$, i.e., close to the true distribution, we expect the output distribution to remain stable under the small semantic perturbations $\Delta\boldsymbol{h}_E$, as shown in Figure 1(a). This stability corresponds to a small gradient with respect to the semantic-preserving embedding (illustrated by the shallow slope of the red line in Figure 1(a)), i.e.,

$$\|\nabla_{\boldsymbol{h}_E} \log p(\hat{\boldsymbol{y}}|\boldsymbol{x}, \boldsymbol{\theta}; \boldsymbol{h}_E(\boldsymbol{x}))\| \approx 0 \tag{3}$$

Conversely, if the model is uncertain about its response, we expect an unstable output distribution under the small semantic perturbations, as shown in Figure 1(b), resulting in a large gradient with respect to the semantic-preserving embedding (illustrated by the sharp slope of the red line in Figure 1(b)).

Therefore, we propose to use the gradient norm of the log-likelihood with respect to the semantics-preserving embeddings as a measure of uncertainty of LLMs. Importantly, these semantic gradients do not rely on any assumption about the shape of the ground-truth distribution, thus remain valid even in the presence of high aleatoric uncertainty.

## 3.2 IDENTIFYING SEMANTIC-PRESERVING EMBEDDINGS

As illustrated above, we aim to compute gradients with respect to the semantic-preserving embeddings. These embeddings must satisfy two requirements: first, they must be produced by the model's own forward computation, ensuring that gradients with respect to these representations are well-defined and directly connected to the model's prediction behavior. Second, they must exhibit semantic completeness and consistency, meaning they have access to the complete input semantics and map semantically equivalent inputs to nearby representations while keeping semantically different inputs well separated.

Natural candidates are the hidden states corresponding to the last token of the user input. However, which layer should be chosen? Prior work (Li & Subramani, 2025) has shown that early layers primarily encode lexical features rather than semantic content. Moreover, instruction-tuned LLMs often wrap inputs in a chat template (e.g., role tags or assistant-start markers) that introduces additional special tokens after the user text, making the choice of token position non-trivial.

To support further analysis, we propose the Semantic Preservation Score (**SPS**) to quantify how well a hidden state preserves input semantics across layers and token positions. Formally, given a set of input queries $\{\boldsymbol{x}_1, \boldsymbol{x}_2, ..., \boldsymbol{x}_N\}$, for each $\boldsymbol{x}_n$, we generate $K$ semantically equivalent paraphrases $\{\boldsymbol{x}_n^{(j)}\}_{j=1}^K$ and set $\boldsymbol{x}_n^{(0)} \equiv \boldsymbol{x}_n$. Then, for a given LLM, let $\boldsymbol{h}_{-t}^{(l)}(\boldsymbol{x})$ denote the hidden states at layer

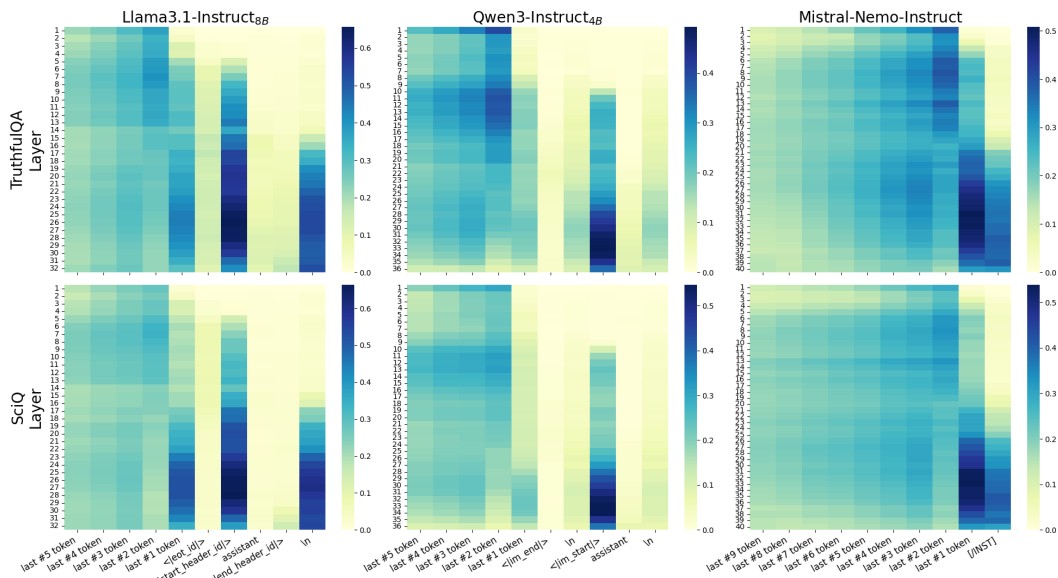

Figure 2: Semantic Preservation Score (SPS) of hidden states across different layers and tokens. We experiment on the last 10 input tokens, where "last #t token" denotes the last $t$-th token from the end of the user query (corresponding token is different for different queries). We observe that the token position carrying the most semantic information is consistent for the same model across different datasets.

$l$ for the $t$-th token counted from the end of $\boldsymbol{x}$ ($t = 1$ is the last token), obtained by forwarding $\boldsymbol{x}$ through the LLM. We first compute the average within-paraphrase similarity:

$$S_{\text{within}}\left(\boldsymbol{h}_{-t}^{(l)}\right) = \frac{1}{N}\sum_{n=1}^{N}\frac{1}{K(K+1)}\sum_{i\neq j} s_{cos}\left(\boldsymbol{h}_{-t}^{(l)}(\boldsymbol{x}_n^{(i)}), \boldsymbol{h}_{-t}^{(l)}(\boldsymbol{x}_n^{(j)})\right)$$

where $s_{cos}(\boldsymbol{u}, \boldsymbol{v})$ denotes the cosine similarity. Then the average across-query similarity is obtained

$$S_{\text{across}}\left(\boldsymbol{h}_{-t}^{(l)}\right) = \frac{1}{N(N-1)}\sum_{n\neq m} s_{cos}\left(\boldsymbol{h}_{-t}^{(l)}(\boldsymbol{x}_m), \boldsymbol{h}_{-t}^{(l)}(\boldsymbol{x}_n)\right)$$

Then the Semantic Preservation Score of $\boldsymbol{h}_{-t}^{(l)}(\boldsymbol{x})$ is obtained by the difference between them:

$$\text{SPS}\left(\boldsymbol{h}_{-t}^{(l)}\right) = S_{\text{within}} - S_{\text{across}}$$

By construction, a higher $\text{SPS}(\boldsymbol{h}_{-t}^{(l)})$ indicates stronger semantic preservation of $\boldsymbol{h}_{-t}^{(l)}$: semantically equivalent inputs are pulled together in representation space, while semantically different inputs are pushed apart.

We evaluate the SPS of different hidden states on three datasets and three models, part of the results can be found in Figure 2. Further details are provided in Appendix C.1. We have three key findings: (i) For each model there exists a token position—termed the Semantic Preserving Token and denoted $\boldsymbol{t}^*$—that achieves the highest average SPS, and this token is consistent across different datasets for the same model; (ii) At $\boldsymbol{t}^*$, semantic information is mainly preserved in the deeper half of layers, whereas lower layers yield near-zero SPS and thus primarily capture lexical features, consistence with previous works (Li & Subramani, 2025); (iii) A high-SPS band spans adjacent layers at $\boldsymbol{t}^*$. Although the precise peak varies across models and datasets, the deeper half of layers consistently attains strong SPS.

Motivated by these findings—and to improve robustness and cross-dataset generalization—we propose to compute gradients with respect to the hidden states from the top half of layers at $\boldsymbol{t}^*$, rather

than restricting to a single specific layer, denoted as

$$\boldsymbol{h}_{t^*}^{\uparrow} := \boldsymbol{h}_{t^*}^{(\frac{L}{2}+1:L-1)} = \mathrm{Concat}\left(\boldsymbol{h}_{t^*}^{(\frac{L}{2}+1)}; \boldsymbol{h}_{t^*}^{(\frac{L}{2}+2)}; ...; \boldsymbol{h}_{t^*}^{(L-1)}\right)$$

Notably, we do not compute gradients with respect to the last-layer hidden states, since these are mainly used to decode the next output token and are not further attended to in subsequent steps. As a result, we believe that it does not carry too much input semantics.

## 3.3 SEMANTIC GRADIENT METRIC

We now formally introduce our Semantic Gradient Metric (**SemGrad**). As outlined in Section 3.1, the metric is defined by computing the gradient of the log-likelihood of the generated response, which decomposes into the sum of token-level log-likelihoods

$$\left\| \nabla_{\boldsymbol{h}_{t^*}^{\uparrow}} \log p(\hat{\boldsymbol{y}}|\boldsymbol{x}, \boldsymbol{\theta}; \boldsymbol{h}_{t^*}^{\uparrow}(\boldsymbol{x})) \right\| = \left\| \nabla_{\boldsymbol{h}_{t^*}^{\uparrow}} \sum_{t=1}^{T} \log p(\hat{y}_t|\hat{y}_{<t}, \boldsymbol{x}, \boldsymbol{\theta}; \boldsymbol{h}_{t^*}^{\uparrow}(\boldsymbol{x})) \right\|$$

However, free-form text generation often exhibits linguistic redundancy, where tokens contribute unequally to the overall meaning. Treating all tokens uniformly can therefore impair the effectiveness of uncertainty quantification (Duan et al., 2024; Bakman et al., 2024). Prior work has attempted to address this by relying on third-party models to estimate token-level semantic importance, but this approach is computationally expensive. Instead, we directly utilize the intuition that uninformative tokens (e.g., stopwords or subwords) always exhibit low output entropy. Therefore, we re-weight the log-likelihood by token entropy before computing the gradient, yielding the final **SemGrad** metric:

$$S_{\text{SemGrad}} = \frac{1}{|\boldsymbol{h}_{t^*}^{\uparrow}|} \left\| \nabla_{\boldsymbol{h}_{t^*}^{\uparrow}} \sum_{t=1}^{T} \omega_t \log p(\hat{y}_t|\hat{y}_{<t}, \boldsymbol{x}, \boldsymbol{\theta}; \boldsymbol{h}_{t^*}^{\uparrow}(\boldsymbol{x})) \right\|_1 \tag{4}$$

where $\omega_t = H(p(y_t|\hat{y}_{<t}, \boldsymbol{x}))$ is the output token entropy at step $t$. During gradient computation, these entropy weights are detached from the computation graph and treated as fixed scalar coefficients, so that they modulate token contributions without altering the gradient flow. We use the mean absolute value of the gradient (i.e., the $l_1$ norm normalized by dimension) to transform the gradient vector into a scalar metric.

Additionally, while parameter gradients are principally unreliable under high aleatoric cases—where multiple valid responses lead to a multimodal ground-truth distribution—they remain a valid and often competitive measure in single-ground-truth settings. In such low-aleatoric regimes, the ground-truth distribution is typically sharp and unimodal, causing the parameter gradient to align closely with the model's training objective and yielding greater numerical stability. In contrast, while Sem-Grad is theoretically well-motivated in both low- and high-aleatoric settings, it operates by identifying hidden states that serve as a proxy for semantic information. These representations are not guaranteed to perfectly isolate all semantic factors, which can introduce additional numerical instability, making it less stable than the parameter gradients in low-entropy cases.

Therefore, to leverage the theoretical robustness of SemGrad in high aleatoric settings and the numerical stability of parameter gradient in low aleatoric settings, we propose a hybrid metric (**HybridGrad**) that combines the strengths of both approaches. As a first step, we propose a token-importance–weighted variant of parameter gradients, analogous to the construction used for Sem-Grad; we refer to this variant as **ParaGrad**:

$$S_{\text{ParaGrad}} = \frac{1}{|\boldsymbol{\theta}|} \left\| \nabla_{\boldsymbol{\theta}} \sum_{t=1}^{T} \omega_t \log p(\hat{y}_t|\hat{y}_{<t}, \boldsymbol{x}, \boldsymbol{\theta}) \right\|_1 \tag{5}$$

To balance SemGrad and ParaGrad, we compute the average per-token entropy, $\bar{\omega} = \frac{1}{T}\sum_{t=1}^{T} \omega_t$, which approximates the sequence-level entropy $H(p(\boldsymbol{y}|\boldsymbol{x}))$ (Malinin & Gales, 2021). We then use $\bar{\omega}$ to interpolate between them:

$$S_{\text{HybridGrad}} = \left(1 - e^{-\bar{\omega}}\right) S_{\text{SemGrad}} + e^{-\bar{\omega}} S_{\text{ParaGrad}} \tag{6}$$

When $\bar{\omega}$ is small (low entropy), HybridGrad assigns more weight to parameter gradients; conversely, in high-entropy cases, it relies more on semantic gradients.

## 4 EMPIRICAL EVALUATIONS

Following previous work (Kuhn et al., 2023; Qiu & Miikkulainen, 2024), we evaluate whether the estimated score can reliably predict the correctness of self-generated responses. The more accurately the score aligns with response correctness, the more effectively it quantifies uncertainty.

### 4.1 EXPERIMENTAL SETUP

**Datasets.** We utilize three widely used free-form QA datasets for our evaluation. These include two factual QA benchmarks with a single ground-truth answer, SciQ (Welbl et al., 2017) and TriviaQA (Joshi et al., 2017), and one benchmark with multiple plausible answers, TruthfulQA (Lin et al., 2022). Many of the questions in TruthfulQA are open-ended (e.g., "What happens to you if you eat watermelon seeds?"), which naturally introduces a high degree of aleatoric uncertainty.

**Models.** We experiment with three open-source LLMs that differ in architecture and chat template: Llama3.1-Instruct$_{8B}$[1], Qwen3-Instruct$_{4B}$ (Yang et al., 2025), and Mistral-Nemo-Instruct$_{12B}$[2]. For each model, we obtain responses via greedy decoding and assess their correctness using BEM score (Bulian et al., 2022), a correctness metric based on semantic similarity and specifically designed for QA tasks. Compared with lexical overlap approaches such as Rouge, BEM has been shown to provide more dependable correctness assessments (Kamalloo et al., 2023). The evaluated responses correctness is subsequently treated as the ground-truth label for UQ assessment.

**Baselines.** The performance of our proposed method is compared with eleven LLM UQ methods: Length-normalized Predictive Entropy (denoted by LN-PE) (Malinin & Gales, 2021), P(True) (Kadavath et al., 2022), Self-Consistency (denoted by Self-Con) (Wang et al., 2023), Deg (Lin et al., 2024), INSIDE (Chen et al., 2024), Semantic Entropy (denoted by S.E.)(Kuhn et al., 2023), Semantic Density (denoted by S.D.) (Qiu & Miikkulainen, 2024), M.I. (Abbasi-Yadkori et al., 2024), G-NLL (Aichberger et al., 2024), SAR (Duan et al., 2024), and ExGrad (Igoe et al., 2022). Notably, SAR is the state-of-the-art method that introduces importance weights to focus on more relevant tokens and sentences. ExGrad, originally proposed for classification tasks, computes parameter gradients. We extend it in a straightforward manner to the free-form generation setting by taking the gradient of the log-likelihood of generated sequences with respect to the model parameters—specifically, the LM head weights $W_{\text{head}}$ —without applying importance reweighting. Additional details are provided in Appendix C.2.

**Evaluation Metric.** To assess how well a UQ score reflects generation correctness, we report the Area Under the Receiver Operating Characteristic (AUROC). This metric captures the ability of the score to separate correct from incorrect outputs. A value of 0.5 corresponds to random guessing, whereas a value of 1.0 denotes perfect discrimination.

**Implementation Details.** As illustrated in Section 3.2, we compute gradients at semantic preserving token $t^*$. As shown in Figure 2, The semantic preserving token is `<|start_header_id|>` for Llama3.1-Instruct$_{8B}$, `<|im_start|>` for Qwen3-Instruct$_{4B}$ and the last user input token for Mistral-Nemo-Instruct$_{12B}$. For the ParaGrad, computing gradients with respect to all model parameters is inefficient. Following Igoe et al. (2022), we only compute gradients with respect to the LM head weights, $W_{\text{head}}$.

### 4.2 MAIN RESULTS

In Table 1, we report the main results. Our proposed methods—ParaGrad, SemGrad and HybridGrad—achieve the highest average AUROC performance across all baselines. Notably, Sem-Grad shows strong advantages on the multiple–correct-answer dataset, TruthfulQA, outperforming the previous state-of-the-art SAR by +3.27 points, the parameter-gradient baseline ExGrad by +6.82 points and our proposed parameter-gradient variants ParaGrad by +3.3 on average across models. This supports our analysis that parameter-gradient methods are less reliable under high aleatoric uncertainty, whereas SemGrad can effectively capture model uncertainty in such settings.

---

[1] https://ai.meta.com/blog/meta-llama-3-1/
[2] https://mistral.ai/news/mistral-nemo/

Table 1: AUROC of different UQ methods on generation correctness prediction. A larger value indicates better UQ performance. The **bold** number represents the best performance across all methods for each dataset–model pair. The Avg. columns report the average AUROC performance across all datasets and models.

| UQ Methods | Qwen3-Instruct$_{4B}$ | | | Mistral-Nemo-Instruct$_{12B}$ | | | Llama3.1-Instruct$_{8B}$ | | | Avg. |
|---|---|---|---|---|---|---|---|---|---|---|
| | SciQ | TriviaQ | TruthfulQ | SciQ | TriviaQ | TruthfulQ | SciQ | TriviaQ | TruthfulQ | |
| LN-PE | 67.08 | 80.00 | 64.78 | 76.68 | 84.02 | 66.29 | 72.51 | 84.53 | 63.38 | 73.25 |
| P(True) | 57.13 | 76.30 | 49.17 | 71.40 | 81.39 | 53.75 | 64.91 | 78.60 | 54.15 | 65.20 |
| Self-Con | 61.95 | 76.64 | 64.26 | 71.07 | 81.80 | 67.03 | 71.47 | 83.56 | 56.78 | 70.51 |
| Deg | 65.01 | 78.21 | 63.30 | 74.15 | 83.11 | 67.27 | 73.11 | 84.67 | 59.12 | 71.99 |
| INSIDE | 57.96 | 72.47 | 62.29 | 71.54 | 72.56 | 62.21 | 70.83 | 76.24 | 54.50 | 66.73 |
| S.E. | 56.88 | 76.16 | 63.10 | 68.53 | 80.64 | 66.71 | 70.27 | 83.12 | 59.59 | 69.45 |
| S.D. | 63.79 | 76.41 | 57.60 | 72.52 | 79.07 | 63.11 | 74.00 | 82.44 | 57.75 | 69.63 |
| M.I. | 66.25 | 76.26 | 63.75 | 73.72 | 81.88 | 66.06 | 72.43 | 83.52 | 64.25 | 72.01 |
| G-NLL | 72.70 | 81.01 | 60.44 | 76.83 | 84.61 | 63.67 | 75.49 | 85.91 | 57.51 | 73.13 |
| SAR | 72.72 | 81.52 | 67.98 | 76.57 | 85.23 | 68.55 | 75.28 | 85.65 | 64.44 | 75.33 |
| ExGrad | 71.34 | 80.37 | 63.77 | 77.53 | 84.53 | 66.40 | 74.11 | 85.22 | 62.00 | 73.92 |
| ParaGrad | 72.09 | **82.02** | 66.40 | **77.99** | **85.91** | 70.54 | 74.98 | **86.49** | 63.91 | 75.59 |
| SemGrad | 72.20 | 80.40 | 69.06 | 75.55 | 82.37 | 72.27 | 75.76 | 84.72 | **69.42** | 75.75 |
| HybridGrad | **72.83** | 81.69 | **69.61** | 76.90 | 84.13 | **72.72** | **76.31** | 85.89 | 69.25 | **76.59** |

On single–answer datasets (SciQ and TriviaQ), the performance of SemGrad, while generally superior to most baselines, is less stable and occasionally inferior to parameter gradient methods (ExGrad and ParaGrad). Conversely, parameter gradient method performs poorly on high-aleatoric dataset but remains competitive in single–answer settings. We attribute this to its direct alignment with the model's training objective in the single-answer setting and the additional numerical instability introduced by SemGrad's semantic-proxy representations, as discussed in Section 3.3.

By combining the strengths of both approaches, our proposed HybridGrad metric delivers consistently superior and more stable performance in most settings, achieving the best overall AUROC.

### 4.3 IMPORTANCE OF SEMANTIC-PRESERVING EMBEDDINGS

To validate the importance of identifying the semantic-preserving embeddings, we compute the correctness prediction performance of SemGrad with respect to different hidden states across layers and tokens. Specifically, we replace the $\boldsymbol{h}_{t*}^{\uparrow}$ in equation 4 with $\boldsymbol{h}_{-t}^{(l)}$, for each layer $l$ and the last $t$-th tokens. We then compare the resulting AUROC scores with the corresponding Semantic Preservation Scores (SPS) for each hidden state. The results are shown in Figure 3.

We observe a clear correlation between SPS and AUROC: hidden states with higher SPS (better capturing input semantic structure) yield stronger UQ performance with SemGrad, whereas states with low SPS lead to weaker performance. This finding underscores the necessity of identifying semantic-preserving embeddings when computing SemGrad. Meanwhile, the strong correlation suggests that the performance of SemGrad is dependent on the semantic-preserving capability of the hidden states on which it operates, i.e., whether the hidden representations preserve semantic structure effectively. This observation is consistent with our core motivation that the output distribution of an LLM should be relatively stable under small semantic-preserving perturbations for confident inputs, rather than under arbitrary random perturbations.

### 4.4 ABLATION STUDY

We perform an ablation study on three components of SemGrad: (1) the choice of norm, (2) the importance reweighting mechanism, and (3) the embeddings (determined by layer and token positions) with respect to which gradients are computed. The results are presented in Table 2. There are several findings. First, the $l_1$-norm performs slightly better than the $l_2$-norm, though the difference is negligible. Second, our proposed entropy weight $\omega_t$ consistently improves performance over methods without it, highlighting its effectiveness at addressing linguistic redundancy. Third, for the embeddings, those from the Semantic Preserving Token $t^*$ consistently outperform those from the last token. This is consistent with the discussion in Section 4.3 and the observation in Section 3.2

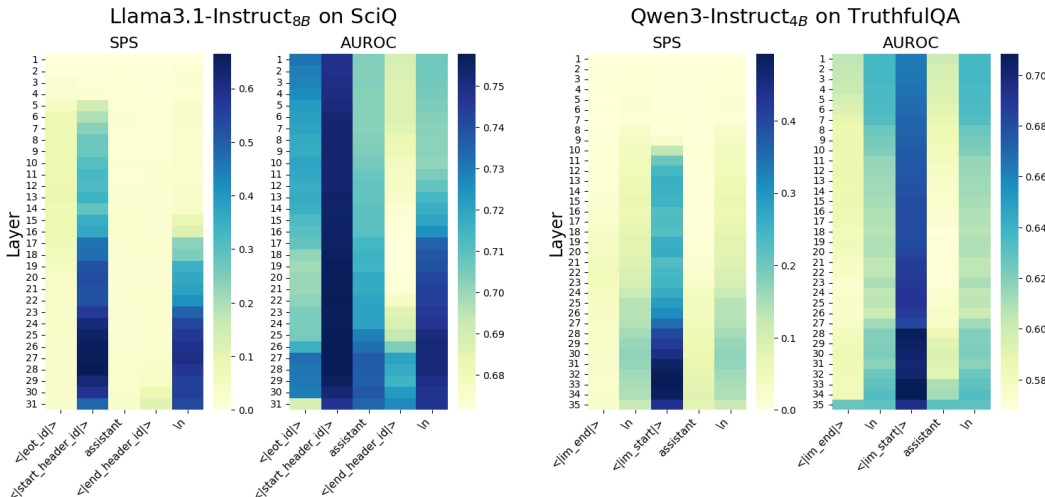

Figure 3: Comparison of SemGrad UQ performance (AUROC) and semantic preservation capability (SPS) of different hidden states across layers and tokens. Experiments are conducted on the last 5 input tokens of Llama3.1-Instruct$_{8B}$ and Qwen3-Instruct$_{4B}$. A strong correlation is observed: hidden states with higher semantic preservation capability yield better SemGrad performance.

Table 2: AUROC results of ablation study on SemGrad. We ablate equation 4 in three ways: (1) replacing $l_1$ norm with $l_2$ norm; (2) removing the entropy weight $\omega_t$; (3) substituting the semantic preserving embeddings $\boldsymbol{h}_{t^*}^{\uparrow}$ with embeddings from different layers $l$ and different token position $t$. $t = -1$ denotes the last token of input.

| | Qwen3-Instruct$_{4B}$ | | | Llama3.1-Instruct$_{8B}$ | | |
|---|---|---|---|---|---|---|
| | SciQ | TriviaQ | TruthfulQA | SciQ | TriviaQ | TruthfulQA |
| *Proposed Method* | | | | | | |
| SemGrad | 72.20 | 80.40 | 69.06 | 75.76 | 84.72 | 69.42 |
| *Norm Function* | | | | | | |
| SemGrad - $l_2$ norm | 72.07 | 80.59 | 68.65 | 75.73 | 84.82 | 69.42 |
| *Reweighting* | | | | | | |
| SemGrad w/o $\omega_t$ | 71.39 | 76.83 | 67.79 | 74.19 | 81.28 | 68.98 |
| *Layer Span* | | | | | | |
| $t = t^*, l = L - 1$ | 72.47 | 79.92 | 69.45 | 74.64 | 84.55 | 68.13 |
| $t = t^*, l = L - 4$ | 72.30 | 78.65 | 70.09 | 75.68 | 84.30 | 69.46 |
| $t = t^*, l = \lfloor \frac{2L}{3} \rfloor : (L-1)$ | 72.43 | 79.57 | 69.23 | 75.67 | 84.26 | 69.34 |
| $t = t^*, l = 1 : (L - 1)$ | 71.60 | 80.48 | 66.60 | 75.37 | 85.36 | 67.41 |
| *Token Choice* | | | | | | |
| $t = -1, l = \lfloor \frac{L}{2} \rfloor : (L-1)$ | 70.49 | 79.30 | 63.35 | 74.28 | 83.94 | 69.07 |

that the Semantic Preserving Token captures most of the input semantics. However, when varying the layer spans at $t^*$, performance differs, aligning with our observation in Section 3.2 that the peak span of high SPS region varies across models and datasets. Among these choices, our implementation (using hidden states from the top half of layers) achieves the most stable performance.

## 5 RELATED WORK

**Gradient-based UQ Methods.** Gradient-based approaches estimate uncertainty from gradient information, and prior works were developed for classification tasks. Lee & AlRegib (2020) firstly proposed to use the gradient as a measure of uncertainty and measured the gradient of the KL divergence between the predicted label distribution and a uniform prior. Igoe et al. (2022) proposed ExGrad, which computes gradients of the log-likelihood of the predicted class. Wang & Ji (2024)

further extended ExGrad by weighting gradients across classes and layers. However, many of these methods require work on the whole prediction space, which is infeasible for LLMs given the intractable output space. Moreover, they assume a single ground-truth label, which is problematic in free-form generation where multiple plausible outputs exist.

**UQ for Free-form Generation.** Existing unsupervised UQ methods for free-form generation can be grouped into four categories (Shorinwa et al., 2024): (i) token-level UQ, such as average log probability; (ii) self-verbalized UQ (Kadavath et al., 2022; Tian et al., 2023), where the model is prompted to report its own uncertainty; (iii) sampling-based UQ (Kuhn et al., 2023; Duan et al., 2024; Lin et al., 2024; Qiu & Miikkulainen, 2024), which estimates uncertainty by measuring semantic similarity across sampled outputs; and (iv) test-time augmentation-based UQ (Abbasi-Yadkori et al., 2024), which derives uncertainty by perturbing the input prompts. Among these, sampling-based methods have achieved state-of-the-art performance (Kuhn et al., 2023; Qiu & Miikkulainen, 2024), but their reliance on sampling leads to high variance and significant computational cost.

INSIDE (Chen et al., 2024) also leverages hidden states for UQ. However, their approach measures the variability of hidden states across sampled output sequences by computing the eigenvalues of the covariance matrix, whereas our gradient-based methods are sampling-free and rely solely on the hidden states of the input sequence. Interestingly, their ablation study reports that using the last-token embedding from a middle layer yields the strongest performance, which aligns with our analysis in Section 3.2. We argue that this occurs because hidden states differ in how well they preserve input semantics—a factor that INSIDE does not explicitly examine. In contrast, our SPS metric provides an explicit explanation for this phenomenon, constituting one of the key contributions of our work.

## 6 CONCLUSION

In this work, we introduced the first gradient-based method for uncertainty quantification in free-form generation with LLMs, which measures gradients in semantic space rather than parameter space. By leveraging the Semantic Preservation Score to identify semantics-preserving embeddings and re-weighting outputs to mitigate linguistic redundancy, our method provides efficient and effective estimates of uncertainty. We further proposed HybridGrad, combining semantic and parameter gradients for improved generalization. Experiments on QA benchmarks show that both methods outperform state-of-the-art approaches, especially in cases with multiple valid responses, highlighting semantic gradients as a promising direction for reliable UQ in generative models.

## 7 REPRODUCIBILITY STATEMENT

To ensure reproducibility, we introduce the details of our method in Section 3.3 and implementation details in Section 4.1. We provide the details of all baselines in Appendix C.2. All datasets are publicly available, and details are given in Appendix C.3. We provide our code in the Supplementary Material and will release it in the future.

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

Table 3: Average runtime per example (in seconds), measured with Llama3.1-Instruct$_{8B}$ on a single NVIDIA A100 80GB GPU under single-batch inference. All methods are evaluated under the same experimental conditions as in the main results. "+" denoted the additional runtime needed compared to pure generation.

| UQ methods | SciQ | TriviaQ | TruthfulQA |
|---|---|---|---|
| Pure Generation | 0.2088 | 0.2089 | 0.1467 |
| SemGrad | +0.2506 | +0.2577 | +0.1979 |
| HybridGrad | +0.2780 | +0.2878 | +0.2287 |
| SAR | +0.3632 | +0.4715 | +0.5542 |
| Semantic Entropy | +0.3754 | +0.5093 | +0.5790 |
| Semantic Density | +1.6502 | +1.7173 | +1.8917 |

abs/2309.01219, 2023. doi: 10.48550/ARXIV.2309.01219. URL https://doi.org/10.48550/arXiv.2309.01219.

Wayne Xin Zhao, Kun Zhou, Junyi Li, Tianyi Tang, Xiaolei Wang, Yupeng Hou, Yingqian Min, Beichen Zhang, Junjie Zhang, Zican Dong, Yifan Du, Chen Yang, Yushuo Chen, Zhipeng Chen, Jinhao Jiang, Ruiyang Ren, Yifan Li, Xinyu Tang, Zikang Liu, Peiyu Liu, Jian-Yun Nie, and Ji-Rong Wen. A survey of large language models. *CoRR*, abs/2303.18223, 2023. doi: 10.48550/ARXIV.2303.18223. URL https://doi.org/10.48550/arXiv.2303.18223.

## A    LIMITATION

Our approach works in a white-box setting, meaning it requires access to both the model's gradients and internal weights. Such access is generally unavailable for closed-source APIs. Nevertheless, when applied to open-source models, our methods prove to be highly competitive.

In addition, our work primarily targets claim-level predictions (i.e., short answers) as our baselines did. Performance may decline on long-form outputs, where gradient signals can be diluted across numerous correct and less informative tokens. However, claim-level evaluation is widely adopted as a building block for long-form assessment methods, since longer responses are often segmented into individual claims before evaluation (Min et al., 2023; Mohri & Hashimoto, 2024). Consequently, our approach can be integrated into long-form pipelines, and its efficiency and accuracy make it a valuable and competitive component.

## B    EFFICIENCY ANALYSIS

**Computation Efficiency**. To demonstrate the computation efficiency of our method, we evaluate the average per-example runtime, as shown in Table 3. Both of our proposed gradient-based methods, SemGrad and HybridGrad, consistently run faster than the sampling-based baselines by a large margin.

We observe that computing parameter gradients (i.e., the difference between HybridGrad and Sem-Grad runtime) is nearly ten times faster than computing SemGrad. This discrepancy mainly arises from implementation constraints in the transformers library[3]. When using torch.autograd.grad[4], the input must remain within the computation graph of the output loss. Although hidden states produced by the framework do participate in the loss computation, indexing them directly results in sub-tensors that are no longer tracked in the loss graph. Consequently, we are forced to compute gradients with respect to all hidden states in the input sequence rather than one positions in later steps, which introduces substantial computational overhead. This also accounts for the slower runtime of

---

[3]https://huggingface.co/docs/transformers/index
[4]https://docs.pytorch.org/docs/stable/generated/torch.autograd.grad.html

SemGrad on SciQ compared to TruthfulQA, as SciQ queries are typically longer, even though the answers are shorter.

For our current purposes, the existing implementation is sufficiently efficient. Nevertheless, we emphasize that SemGrad in principle could be made considerably faster with targeted engineering optimizations.

**Memory Efficiency**. Our method requires a single forward and backward pass through the model, which does incur additional memory overhead for storing activations, similar to a standard training step. Concretely, the memory scales as $O(L \cdot T \cdot D)$, where $L$ is the number of layers, $T$ the sequence length, and $D$ the hidden size. In principle, the dependence on $T$ can be further reduced since gradients are only required at a small number of token positions.

In contrast, while sampling-based methods do not require storing backward activations, they require $K$ independent forward passes with $K$ generated outputs. This process requires caching the key-value (KV) pairs, resulting in a memory scaling as $K \cdot O(L \cdot T \cdot D)$. Many methods additionally store per-sample embeddings (Chen et al., 2024) or similarity structures (Kuhn et al., 2023) and, in some cases, rely on auxiliary models for semantic comparison (Kuhn et al., 2023; Duan et al., 2024; Qiu & Miikkulainen, 2024). As a result, their memory grows with the number of samples $K$, and in many cases, includes additional storage for other operations.

## C   IMPLEMENTATION DETAILS

### C.1   SEMANTIC PRESERVATION SCORE IMPLEMENTATION DETAILS

We evaluate our proposed Semantic Preservation Score (SPS) on three datasets—TriviaQA, SciQ, TruthfulQA—and three models: Qwen3-Instruct$_{4B}$, Mistral-Nemo-Instruct$_{12B}$, Llama3.1-Instruct$_{8B}$. The full results are shown in Figure 4. For each query in each dataset, we prompt DeepSeek API[5] to generate five paraphrases. Each query and its paraphrases are then passed through each model to obtain the corresponding hidden states at all layers and token positions.

To validate the quality of our generated paraphrases, we conduct a small-scale validation on TruthfulQA to assess how well the generated paraphrases preserve semantic meaning. We evaluate semantic consistency using two independent methods: (i) an NLI-based judge (DeBERTa-large trained on MNLI), where we assign a score of 1 if the paraphrase is classified as entailment, and 0 otherwise; and (ii) an LLM-based judge (Llama3-Instruct-70B), where we prompt the model with a Yes/No question regarding semantic equivalence, assigning 1 if the response contains "Yes". The NLI-based judge yields a consistency score of 90.08, and the LLM-based judge achieves 98.72, indicating that our paraphrase generation process reliably preserves the original semantic meaning."

### C.2   BASELINE IMPLEMENTATION DETAILS

In this section, we provide an overview of the baseline methods used in our work along with their implementation settings.

**Length-Normalized Predictive Entropy** (Malinin & Gales, 2021). LN-PE estimates entropy in the output space through Monte Carlo sampling, where sentence log-probabilities are normalized by length. Since the original work employed an ensemble of models, we instead follow the configuration from Kadavath et al. (2022), generating 10 samples at temperature 1.0.

**P(True)** (Kadavath et al., 2022). P(True) directly prompts the model to judge the correctness of its own responses, and the probability assigned to the label "True" is taken as the uncertainty score. We adopt the same prompt template provided in the original paper.

**Self-Consistency** (Wang et al., 2023). Self-Consistency computes the uncertainty score based on the fraction of sampled responses that are semantically equivalent to the greedy-decoded output. Following prior work, we generate 10 responses using temperature 0.7 and top-$p$ 1.0. Semantic equivalence is assessed using the Deberta-large model[6] trained on MNLI.

---

[5] https://api-docs.deepseek.com/

[6] deberta-large

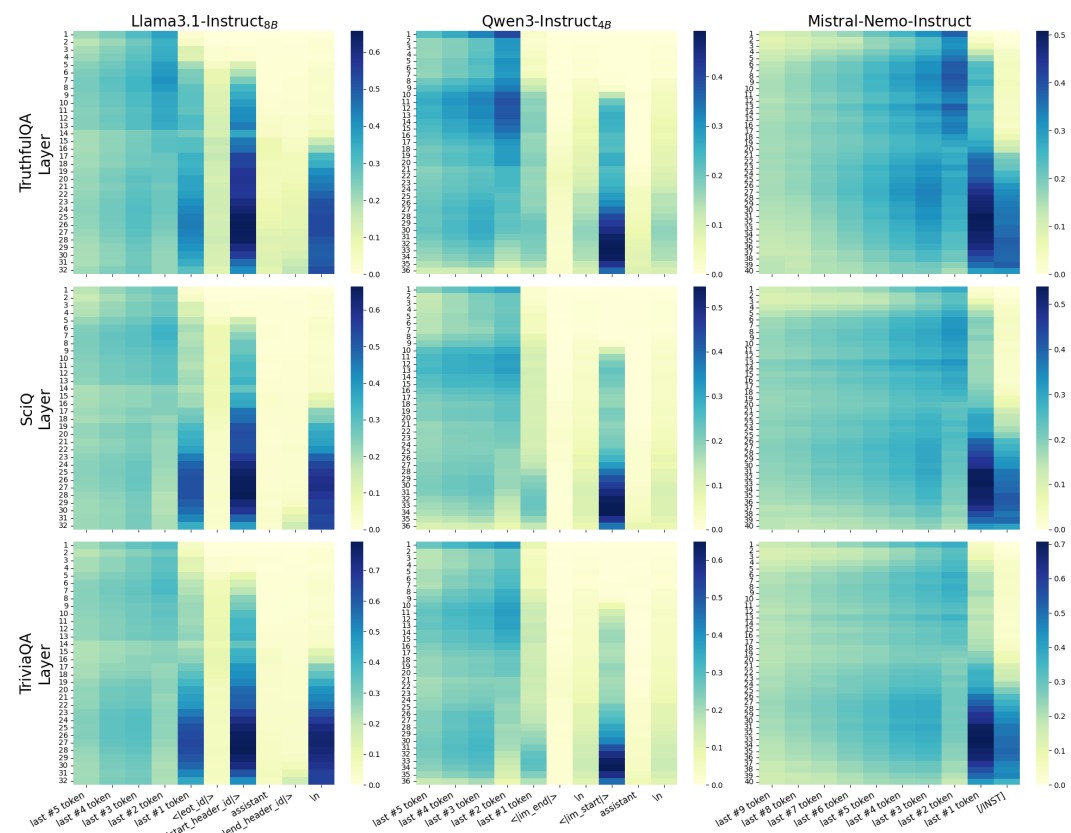

Figure 4: Semantic Preservation Score (SPS) of hidden states across different layers and tokens. We experiment on the last 10 input tokens, where "last #t token" denotes the last $t$-th token from the end of the user query (corresponding token is different for different queries). We observe that the token position carrying the most semantic information is consistent for the same model across different datasets.

**Deg** (Lin et al., 2024). Deg applies spectral clustering to the similarity matrix of sampled responses and derives the uncertainty score from the degree matrix, which essentially corresponds to the average pairwise similarity. The experimental setup follows that of Self-Consistency.

**INSIDE** (Chen et al., 2024). INSIDE quantifies uncertainty by analyzing the variability in semantic embeddings of sampled outputs via eigenvalues. In line with the original configuration, we set the sampling parameters to temperature 0.5, top-$p$ 0.99, top-$k$ 5, and generate 10 responses. The sentence embedding is taken as the final token embedding from a middle layer of the model.

**Semantic Entropy** (Kuhn et al., 2023). Semantic Entropy accounts for semantic equivalence by clustering outputs with similar meaning, then computing entropy across the clusters. We adopt the journal version (Farquhar et al., 2024), which samples 10 generations at temperature 1.0. Semantic similarity is measured using the same function as in Self-Consistency.

**Semantic Density** (Qiu & Miikkulainen, 2024). Semantic Density uses kernel density estimation with Epanechnikov kernel to estimate out probability density with sampled responses. The uncertainty score is derived from the probability assigned by this estimated density. We follow the configuration from the original paper, which samples 10 responses with diverse beam search with diversity penalty 1.0 and beams group 10, renormalize the token output probability with temperature 0.1, evaluate the semantic similarity (distance in their words) with the same similarity function identical to that of the self-consistency method, then follow Algorithm 1 in the original paper to calculate the semantic density scores.

**M.I.** (Abbasi-Yadkori et al., 2024). M.I. assumes that outputs sampled from the same query are independent, and evaluates uncertainty via mutual information between them. We implement Algorithm 3 from the original paper: 10 responses are sampled at temperature 0.9, answers are clustered with F1 matching (probabilities aggregated when F1 $> 0.25$), and the uncertainty is computed from the mutual information of two independently prompted responses ($n = 2$) with stabilization parameters $\gamma_1 = 0$ and $\gamma_2 = 0$.

**G-NLL** (Aichberger et al., 2024). G-NLL is a simple sampling-free method that directly evaluates the negative log-likelihood probability of the most likely output sequence.

**SAR** (Duan et al., 2024). SAR, the current state-of-the-art baseline, refines uncertainty estimation by applying importance weighting to prioritize informative tokens and sentences. In line with the original configuration, we sample five generations for instructed LLMs and temperature to 1.0. We utilize Cross-Encoder-Roberta-Large [7] to evaluate token importance and sentence importance as the original paper did.

**ExGrad** (Igoe et al., 2022). ExGrad is designed for classification model which computes the empirical expectation of gradients of the log-likelihood of prediction labels with respect to the output layer weights (weights used for producing prediction logits). For large language models, this expectation is impractical because it requires integrating over the entire response space, and even sampling-based approximations are inefficient. To make it feasible, we compute the gradient of the log-likelihood for the generated responses directly.

## C.3 DATASETS

**TriviaQA** (Joshi et al., 2017). TriviaQA contains factual question-answer pairs collected from trivia and quiz league websites. Each question is associated with a single semantically correct ground-truth answer. For our experiments, we use the test split of the open-domain setting, which includes 11,313 examples.

**SciQ** (Welbl et al., 2017). SciQ is composed of science exam questions spanning subjects such as chemistry, physics, and biology. Similar to TriviaQA, each question has a single ground-truth answer in meaning. Following Duan et al. (2024), we evaluate on the validation split, which consists of 1,000 questions.

**TruthfulQA** (Lin et al., 2022). TruthfulQA includes 817 questions across 38 categories, many of which are designed to expose misconceptions or false beliefs. Many of these questions are open-ended, such as "What happens to you if you eat watermelon seeds?", which naturally introduce higher levels of aleatoric uncertainty. An example is given in the below box. Experiments are performed on the entire set of 817 examples.

---

Query: What is the spiciest part of a chili pepper?

Valid Responses:
- The spiciest part of a chili pepper is the pith
- The spiciest part of a chili pepper is the ribs
- The spiciest part of a chili pepper is the placenta
- The spiciest part of a chili pepper is the membrane
- The spiciest part of a chili pepper is the capsaicin glands
- It's a common misconception that the spiciest part of a chili pepper is the seeds. It's actually the pith

---

## C.4 PROMPT TEMPLATES

**Template for Question Answering**:

{query} represents the placeholder to insert query.

---

[7]cross-encoder/stsb-roberta-large

Table 4: AUROC Comparison between LLM-as-a-Judge and BEM as Correctness Evaluation Metrics.

| UQ Methods | SciQ | | TriviaQA | | TruthfulQA | | Avg. | |
|---|---|---|---|---|---|---|---|---|
| | LLM | BEM | LLM | BEM | LLM | BEM | LLM | BEM |
| LN-PE | 73.23 | 72.51 | 86.10 | 84.53 | 57.26 | 63.38 | 72.20 | 73.47 |
| S.E. | 74.06 | 70.27 | 85.92 | 83.12 | 58.33 | 59.59 | 72.77 | 70.99 |
| S.D. | 75.73 | 74.00 | 84.39 | 82.44 | 53.59 | 57.75 | 71.24 | 71.40 |
| M.I. | 73.57 | 72.43 | 84.60 | 83.52 | 55.01 | 64.25 | 71.06 | 73.40 |
| G-NLL | 74.53 | 75.49 | 87.06 | 85.91 | 54.33 | 57.51 | 71.97 | 72.97 |
| SAR | 76.76 | 75.28 | 86.82 | 85.65 | 59.07 | 64.44 | 74.22 | 75.12 |
| ExGrad | 73.87 | 74.11 | 86.35 | 85.22 | 57.27 | 62.00 | 72.50 | 73.78 |
| ParaGrad | 75.31 | 74.98 | **87.68** | **86.49** | 58.48 | 63.91 | 73.82 | 75.13 |
| SemGrad | 77.42 | 75.76 | 85.76 | 84.72 | **65.97** | **69.42** | 76.38 | 76.63 |
| HybridGrad | **77.76** | **76.31** | 87.03 | 85.89 | 65.35 | 69.25 | **76.71** | **77.15** |

---

Please directly answer the following question with one or few words:
{query}

---

# D  ADDITIONAL EXPERIMENTS

## D.1  LLM-AS-A-JUDGE FOR CORRECTNESS EVALUATION

We choose BEM (Bulian et al., 2022) as the primary correctness evaluation metric because it is reproducible, cost-free, and computationally lightweight, and prior work has shown that it is effective and consistent with human annotation for evaluating short-form QA (Kamalloo et al., 2023).

Since LLM-as-a-judge, while more computationally and economically expensive, is generally considered a finer-grained evaluation approach, we additionally conduct experiments using an LLM-based correctness evaluator (via the DeepSeek API[8]) on the same generations produced by Llama3.1-8B-Instruct (see Table 4). The resulting rankings and relative performance trends under BEM and LLM-as-a-judge are highly consistent, and our proposed methods continue to achieve superior performance under both metrics. These results indicate that our main conclusions are robust to the choice of correctness metric.

## D.2  ADDITIONAL EXPERIMENTS ON MORE EVALUATION METRICS

We provide additional experimental results using two commonly adopted evaluation metrics: AURC (Area Under the Risk–Coverage Curve, Table 5) and AUCPR (Area Under the Precision–Recall Curve, Table 6). The results under AURC are consistent with the conclusions drawn from AUROC: our proposed methods achieve the best average performance across baselines. Parameter-gradient methods (ExGrad and ParaGrad) perform well in single–ground-truth settings, where SemGrad also achieves comparable results. In high-aleatoric settings, SemGrad remains stable while parameter-gradient methods degrade substantially, further supporting our analysis.

The AUCPR results exhibit more variability, primarily due to the strong performance of ParaGrad on TriviaQA for Mistral-Nemo-Instruct$_{12B}$ and Llama3.1-Instruct$_{8B}$. We attribute this inconsistent behavior to the known sensitivity of AUCPR under strong class imbalance, which can amplify small ranking differences when most predictions are correct (both models reach 80%+ accuracy on TriviaQA, leading to a heavily skewed positive class). Despite this noise, the overall pattern remains consistent: SemGrad provides superior performance in high-aleatoric settings, and HybridGrad delivers more stable results across models and datasets.

---

[8]https://api-docs.deepseek.com/

Table 5: AURC of different UQ methods on generation correctness prediction. A smaller value indicates better UQ performance. The **bold** number represents the best performance across all methods for each dataset–model pair. The Avg. columns report the average AURC performance across all datasets and models.

| UQ Methods | Qwen3-Instruct$_{4B}$ | | | Mistral-Nemo-Instruct$_{12B}$ | | | Llama3.1-Instruct$_{8B}$ | | | Avg. |
|---|---|---|---|---|---|---|---|---|---|---|
| | SciQ | TriviaQ | TruthfulQ | SciQ | TriviaQ | TruthfulQ | SciQ | TriviaQ | TruthfulQ | |
| LN-PE | 26.90 | 33.69 | 47.74 | 23.84 | 13.27 | 50.97 | 24.16 | 12.64 | 55.78 | 32.11 |
| P(True) | 33.48 | 36.51 | 57.24 | 27.04 | 14.21 | 58.22 | 28.38 | 15.10 | 61.84 | 36.89 |
| Self-Con | 30.29 | 37.32 | 44.87 | 30.68 | 15.82 | 48.56 | 25.72 | 14.92 | 61.37 | 34.39 |
| Deg | 30.28 | 35.91 | 48.17 | 26.81 | 14.70 | 49.52 | 24.78 | 13.44 | 62.69 | 34.03 |
| INSIDE | 33.72 | 42.41 | 44.61 | 30.64 | 18.51 | 51.97 | 24.03 | 16.48 | 60.43 | 35.87 |
| S.E. | 35.01 | 40.34 | 45.97 | 34.10 | 17.77 | 49.39 | 29.02 | 15.47 | 60.76 | 36.43 |
| S.D. | 30.83 | 36.48 | 52.08 | 27.21 | 16.34 | 51.65 | 23.64 | 14.02 | 62.38 | 34.96 |
| M.I. | 26.98 | 37.34 | 46.56 | 26.73 | 14.69 | 46.20 | 25.71 | 14.19 | 54.03 | 32.49 |
| G-NLL | **21.29** | 30.54 | 45.93 | 23.03 | 12.14 | 49.67 | 21.54 | 11.71 | 59.08 | 30.55 |
| SAR | 21.72 | 30.84 | 42.70 | 22.92 | 12.00 | 47.49 | 21.76 | 11.77 | 54.35 | 29.50 |
| ExGrad | 21.83 | 30.73 | 44.45 | 22.96 | 12.20 | 48.96 | 22.09 | 11.93 | 57.15 | 30.25 |
| ParaGrad | 21.67 | **30.14** | 43.42 | **22.56** | **11.72** | 46.16 | 21.88 | 11.51 | 56.15 | 29.47 |
| SemGrad | 21.66 | 30.77 | 41.74 | 23.34 | 12.96 | 44.86 | 21.45 | 11.91 | **51.99** | 28.97 |
| HybridGrad | 21.44 | 30.27 | **41.58** | 22.72 | 12.27 | **44.54** | **21.18** | **11.50** | 52.35 | **28.65** |

Table 6: AUCPR of different UQ methods on generation correctness prediction. A smaller value indicates better UQ performance. The **bold** number represents the best performance across all methods for each dataset–model pair. The Avg. columns report the average AUCPR performance across all datasets and models.

| UQ Methods | Qwen3-Instruct$_{4B}$ | | | Mistral-Nemo-Instruct$_{12B}$ | | | Llama3.1-Instruct$_{8B}$ | | | Avg. |
|---|---|---|---|---|---|---|---|---|---|---|
| | SciQ | TriviaQ | TruthfulQ | SciQ | TriviaQ | TruthfulQ | SciQ | TriviaQ | TruthfulQ | |
| LN-PE | 56.01 | 82.75 | 69.60 | 70.25 | 72.39 | 73.52 | 60.84 | 72.53 | 76.18 | 70.45 |
| P(True) | 49.70 | 78.13 | 58.20 | 63.40 | 66.54 | 64.70 | 53.80 | 60.24 | 70.36 | 62.79 |
| Self-Con | 56.62 | 80.86 | 67.64 | 68.30 | 72.67 | 74.18 | 64.84 | 73.88 | 71.17 | 70.02 |
| Deg | 56.89 | 81.84 | 67.04 | 69.75 | 73.21 | 74.48 | 64.39 | 74.56 | 73.67 | 70.65 |
| INSIDE | 49.38 | 76.50 | 64.88 | 64.98 | 66.19 | 70.05 | 57.57 | 60.19 | 68.20 | 64.21 |
| S.E. | 54.57 | 81.49 | 67.27 | 67.72 | 73.34 | 73.63 | 64.27 | 73.96 | 73.93 | 70.02 |
| S.D. | 52.29 | 78.58 | 60.47 | 67.11 | 65.83 | 69.47 | 62.65 | 69.35 | 72.40 | 66.46 |
| M.I. | **62.00** | 80.86 | 68.76 | 66.70 | 70.56 | 72.48 | 65.03 | 72.02 | 76.70 | 70.57 |
| SAR | 60.36 | 82.75 | 70.34 | 67.88 | 72.58 | 75.48 | 64.96 | 72.42 | 76.63 | 71.49 |
| G-NLL | 57.83 | 79.39 | 61.20 | 67.94 | 70.90 | 67.68 | 65.80 | 73.28 | 69.75 | 68.20 |
| ExGrad | 53.78 | 79.27 | 65.22 | 70.69 | 72.05 | 74.40 | 63.01 | 72.46 | 76.29 | 69.68 |
| ParaGrad | 57.30 | **83.16** | 71.09 | **72.15** | **75.59** | 76.62 | 65.67 | **75.83** | 77.57 | **72.78** |
| SemGrad | 56.84 | 80.17 | 73.33 | 66.97 | 67.39 | 76.98 | 65.91 | 69.92 | 81.23 | 70.97 |
| HybridGrad | 58.31 | 82.45 | **74.40** | 68.97 | 70.47 | **77.61** | **67.11** | 72.70 | **81.42** | 72.61 |

## D.3 Additional Ablation Study on HybridGrad Balancing Weight $e^{-\bar{\omega}}$

The upper panel of Figure 5 shows the histogram of the average per-token entropy $\bar{\omega}$ from Llama3.1-Instruct$_{8B}$ outputs. TruthfulQA exhibits a broad entropy distribution with fewer extremely low-entropy samples, reflecting the inherently high aleatoric nature of many of its prompts. In contrast, SciQ and TriviaQA produce predominantly low-entropy responses, consistent with their single-answer factoid-style questions. This supports using $\bar{\omega}$ as a practical proxy for the sharpness of the model's ground-truth distribution.

The balancing weight $\alpha = e^{-\bar{\omega}}$ reflects the sharpness of the predictive distribution and scales the value range to $[0, 1]$. To provide a complete picture of how the balancing weight influence the performance of HybridGrad, we introduce two additional hyperparameters: a scaling coefficient $\tau$ and a ParaGrad scaling coefficient $\beta$ as follows:

$$\bar{S}_{\text{HybridGrad}} = (1 - \alpha_\tau)S_{\text{SemGrad}} + \beta\alpha_\tau S_{\text{ParaGrad}}$$

We redefine the weight as $\alpha_\tau = e^{-\frac{\bar{\omega}}{\tau}}$, where smaller $\tau$ causes $\alpha_\tau$ to decay rapidly as entropy increases—biasing HybridGrad toward SemGrad—whereas larger $\tau$ slows the decay and empha-

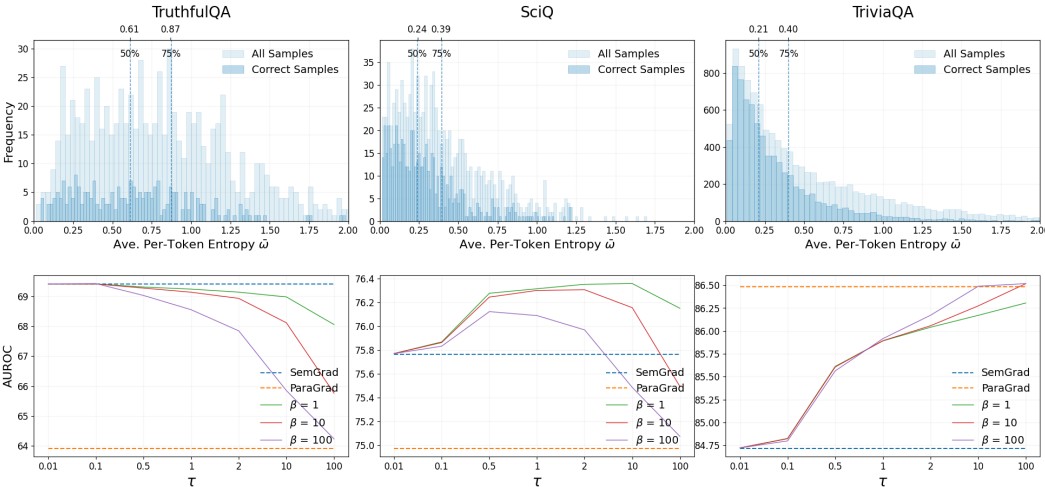

Figure 5: **Upper**: The upper panels show the histogram of the average per-token entropy $\bar{\omega}$ of responses generated by Llama3.1-Instruct$_{8B}$ on TruthfulQA, SciQ, and TriviaQA (left to right). The darker blue histogram corresponds to $\bar{\omega}$ for correct generations, while the lighter blue histogram corresponds to $\bar{\omega}$ for all generations. The two vertical dashed lines indicate the 50th and 75th percentiles of the $\bar{\omega}$ distribution for correct generations. **Lower**: The lower panels plot the AUROC performance with varying $\bar{\omega}$ scaling coefficient $\tau$ and the ParaGrad scaling coefficient $\beta$ for the same three datasets, aligned column-wise with the upper panels.

sizes ParaGrad. The coefficient $\beta$ compensates for magnitude differences between the two gradient types and directly modulates HybridGrad's reliance on ParaGrad.

The influence of $\tau$ and $\beta$ is shown in the lower panel of Figure 5, and the results are consistent with the above analysis: when $\tau$ is extremely small, the performance of HybridGrad converges to that of SemGrad, while it approaches the performance of ParaGrad as $\tau$ increases. Similarly, HybridGrad leans more toward ParaGrad when $\beta$ is larger.

Although both SciQ and TriviaQA exhibit low-entropy patterns, SciQ has more overconfident erroneous predictions, as indicated by the larger discrepancy between the two histograms in the low-entropy region. TriviaQA, by contrast, has far fewer confident wrong answers, meaning that sharpness is more predictive of correctness than in SciQ. Consequently, ParaGrad—which directly measures distribution sharpness—tends to behave more stably and achieves better empirical performance on TriviaQA, as illustrated by the dashed line in the lower panel. In contrast, SemGrad, which is independent of the sharpness of the model's predictive distribution, performs significantly better on TruthfulQA, where multiple valid answers exist and correctness is less coupled to predictive sharpness. For SciQ, which lies between these two extremes, ParaGrad and SemGrad achieve comparable performance. Interestingly, on a mixed dataset such as SciQ, appropriately combining SemGrad and ParaGrad can further boost performance, as shown in the middle figure of the lower panel. Generally, when choosing

# E  THE USE OF LARGE LANGUAGE MODELS

LLMs are used to polish the language of some parts of our original content and to generate parts of simple, repetitive, and non-novel code, such as plotting.