# OpenReview forum: "Gradients with Respect to Semantics Preserving Embeddings Tell the Uncertainty of Large Language Models"
_ICLR.cc/2026/Conference — Submitted to ICLR 2026_

### Official Review · Reviewer_QuTT · 2025-10-17

**Soundness:** 2
**Presentation:** 3
**Contribution:** 3
**Rating:** 4
**Confidence:** 5

**Summary:**

This work introduces SemGrad, which utilizes gradient information in semantic space for uncertainty estimation for hallucination detection.
To that end, it further introduces the SPS to identify embeddings that cary semantic information.
Finally, the paper introduces a hybrid approach - HybridGrad - that interpolates between SemGrad and gradient information w.r.t. the parameters, which I will refer to for this review as ParamGrad for simplicity. Both methods are evaluated across multiple benchmark datasets and LLMs.

**Strengths:**

- The paper is well organized and clearly written.
- The idea of using gradient information of the LLM for hallucination detection is a straightforward idea and the extention to gradient w.r.t. semantic embeddings is an interesting and novel idea.
- A good selection of baselines methods on three LLMs in the 7B range.
- Very interesting investigation with SPS, analyzing the specifics of the different used LLMs regarding what layer and token is most useful to extract semantic information. This is useful even for competitor methods like INSIDE that rely on embedding information.
- Thoughtful ablation studies, rationalizing many of the choices for implementing the method.

**Weaknesses:**

- The BEM evaluation metric should be complimented with LLM-as-a-judge (Zheng et al.) as correctness metric, which has emerged as the de-facto standard for correctness evaluation for hallucination detection.
- AUROC is widely used as an evaluation metric in those experiments, other metrics such as PRR (Malinin & Gales) would provide a more complete picture.
- I find the deduction that SemGrad operates in semantic space (line 417) not entirely convincing, I either suggest toning down the claim or provide more convincing evidence.
- The motivation from using the parameter gradient norm is understandable, albeit a bit weak. I agree that at the true model it has the correct behavior is expressing complete certainty, yet there is no guarantee on monotonicity or am I mistaken? That is, why do we know that one prediction with a higher gradient norm than another is more uncertain? All the argument around Eq.(1) and (2) tells us is, that under true model parameters the gradient norm will be zero.

Notes:
- Equation references should use \eqref
- I would prefer a single-color colormap for figures 2-4, the used one suggests correlated - neutral - anticorrelated
- The findings of the usefulness of embeddings within the layers appears strongly related to the findings of Chen et al. 2024, which would be worthwile to discuss.
- Shouldn't line 418 state "semantic *preserving* perturbations"?

---
Zheng, Chiang, ... (2023) Judging LLM-as-a-Judge with MT-Bench and Chatbot Arena, NeurIPS

Malinin, Gales (2021) Uncertainty Estimation in Autoregressive Structured Prediction, ICLR

Chen, liu, ... (2024) INSIDE: LLMs' Internal States Retain the Power of Hallucination Detection, ICLR

**Questions:**

- What is the performance of ParamGrad alone? SemGrad and HybridGrad have been evaluated in the main experiments in Table 1, yet not the ParamGrad component. I know that ExGrad is similar, yet lacking the token-entropy weighting which would be interesting to evaluate as a standalone.
- I wonder about the justification of adding up SemGrad and ParamGrad to form HybridGrad, I would like to see a comparison if they are usually within similar ranges. For me it is not clear that they should be, as gradient norms in parameter space and semantic embedding space might be in completely different ballparks. Could one also consider a multiplicative
- I would ask for an elaboration on when the particular weighting between SemGrad and ParamGrad in Eq.(5) is expected to work. What I really like about it is its input dependency, yet the exact values for the entropies depend on a lot of things that are not particularly discussed in the present work. For example different vocabulary sizes or different precisions or stability tricks for calculating the next token distribution could change it drastically. In my opinion, there should be an explicit temperature hyperparameter (i.e. $e^{-\bar{w}/\tau}$) making this dependency more verbose and allowing for systematic study of the impact of this weighting.
- Do you observe any length-dependency in the metrics and if yes how strong? Would there be any merit in length-normalization as widely used in prior work, e.g. in many of the discussed baselines, or in using the weighting introduced in the work on SAR?
- Are the negative answers featured in the TruthfulQA dataset used for evaluation?
- How are the semantically equivalent paraphrases generated? Was there any investigation into how well this additional generated sequences preserve semantics?
- Apart from increased runtime, does the weighting based on thrid party models as in e.g. Duan et al. further improve the method compared to the token-entropy based weighting?
- What codebase was used to estimate semantic entropy? The original one from Kuhn et al. or the corrected estimator from Aichberger et al.? Note that the codebase from the Nature version of Semantic entropy (Farquhar et al.) also uses this corrected estimator.
- I would further include just PE and also LN-SE to disentangle the effect of length-normalization, or is there a reason why this was not included?

---
Aichberger, Schweighofer, ... (2024) Improving Uncertainty Estimation through Semantically Diverse Language Generation, ICLR

Farquhar, Kossen, ... (2024) Detecting hallucinations in large language models using semantic entropy, Nature

---

> ### Author Response · Authors · 2025-11-21
> **Author Response I**
>
> Thank you for your valuable feedback and constructive suggestions. We appreciate the reviewer’s careful evaluation and acknowledgment of our work. Below is our response to your comments, which we hope will answer your questions and concerns.
>
> > ### ***1. Response to Correctness Metric BEM (Weakness 1)***
>
> Thank you for your valuable feedback. We fully agree that LLM-as-a-judge can provide more fine-grained judgments and can complement our evaluation to improve robustness. We chose BEM because it is ***computationally lightweight, reproducible, and cost-free***, and prior work has shown it to be an effective metric for evaluating short-form QA answers.
>
> Specifically, we followed prior work by Kamalloo et al. (2023)[1], which systematically compared automatic metrics (including LLM-as-judge) against human annotations, and here is their conclusion in the introduction:
>
> > "automated evaluation mechanisms such as BEM based on semantic matching between the gold answers and generated answers produce a relative performance that is mostly consistent with human evaluation."
>
> This conclusion does not hold for long-form generation. However, our work focuses on claim-level, short-answer settings (as discussed in the Limitations section) and, therefore, aligns with the above conclusion.
>
> To address your concern, we conducted additional experiments using an LLM-as-a-judge (with DeepSeek API) evaluator on the same generations from Llama3.1-8B-Instruct. The results, compared with the BEM-based evaluation, are shown below:
>
> | UQ Method   | SciQ |  | TriviaQ  |  | TruthfulQ  |   | Avg. |  |
> |------------|------------|------------|---------------|---------------|----------------|----------------|------|--|
> |   | LLM |  BEM |  LLM | BEM |  LLM |  BEM | LLM  | BEM  |
> | LN-PE      | 73.23 | 72.51 | 86.10 | 84.53 | 57.26 | 63.38 |  72.20 | 73.47|
> | S.E.       | 74.06 | 70.27 | 85.92 | 83.12 | 58.33 | 59.59 |  72.77 | 70.99|
> | S.D.       | 75.73 | 74.00 | 84.39 | 82.44 | 53.59 | 57.75 | 71.24 | 71.40|
> | M.I.       | 73.57 | 72.43 | 84.60 | 83.52 | 55.01 | 64.25 | 71.06 | 73.40|
> | SAR    | 76.76 | 75.28 | 86.82 | 85.65 | 59.07 | 64.44 | 74.22 | 75.12|
> | ExGrad     | 73.87 | 74.11 | 86.35 | 85.22 | 57.27 | 62.00 | 72.50| 73.78|
> | SemGrad | 77.42 | 75.76 | 85.76 | 84.72 | **65.97** | **69.42** | 76.38| 76.63|
> | HybridGrad | **77.76** | **76.31** | **87.03** | **85.89** | 65.35 | 69.25 | ***76.71***| ***77.15***|
>
> As shown, the rankings and relative trends under BEM and LLM-as-a-judge are highly consistent, and our proposed methods continue to achieve superior performance under both metrics. This suggests that our main conclusions are robust to the choice of correctness metric. We will add this additional experiment to the Appendix.
>
> [1] Kamalloo et al. (2023). Evaluating open-domain question answering in the era of large language models. ACL 2023.

---

> > ### Author Response · Authors · 2025-11-21
> > **Author Response II**
> >
> > > ### ***2. Response to More Evaluation Metrics (Weakness 2)***
> >
> > Thank you for your valuable suggestion. We add experiments with additional metrics, including: AUCPR (Area Under the Precision–Recall Curve), AURC (Area Under the Risk–Coverage Curve), SA@X (Selective Accuracy at X% Coverage), and PRR (prediction rejection area ratio PRR from Malinin & Gales).
> >
> > We provide the results on TruthfulQA and SciQ with model Llama3.1-8b-Instruct as follows:
> >
> > | UQ Method   | SciQ  |  ||  |  |TruthfulQ  |  ||  |  |
> > |------------|------------|------------|---------------|---------------|----------------|------------|------------|---------------|---------------|----------------|
> > |   |  AUROC | AUCPR | AURC | PRR |  SA@90 | AUROC |  AUCPR |AURC |  PRR | SA@90 |
> > | LN-PE      | 72.51|60.84|24.16|-0.41|63.22|63.38|76.18|55.78|-5.62|37.01|
> > | S.E.       | 70.27|64.27|29.02|-0.86|64.11|59.59|73.93|60.76|-6.83|36.19|
> > | S.D.       | 74.00|62.65|23.64|-0.36|63.56|57.75| 72.40|62.38|-7.23|36.19|
> > | M.I.       | 72.43|65.03|25.71|-0.55|***65.67***|64.25|76.70|54.03|-5.20|37.28|
> > | SAR    | 75.28|64.96|21.75|-0.18|64.44|64.44|76.63|54.35|-5.27|36.73|
> > | ExGrad     |74.11|63.01|22.09|-0.21|63.67|62.04|76.29|57.15|-5.96|37.28|
> > | **SemGrad** |75.76|65.91|21.45|-0.15|64.67| ***69.42***|81.23|***51.99***|***-4.70***|37.82|
> > | **HybridGrad** |***76.31***|***67.11***|***21.18***|***-0.13***|64.78| 69.25|***81.42***|52.35|-4.79|***37.96***|
> >
> > Our methods still consistently outperform all baselines across most settings. We will add this experiment to the Appendix. Thanks again for your suggestion.
> >
> > We do not find an official implementation of PRR and therefore implemented it based on the definition in the original paper. We think our implementation is correct, but the resulting score is indeed strange (which is negative). However, we would like to argue that PRR is an affine decreasing function of AURC, so reporting one of them is sufficient.
> >
> > Note that PRR is defined as:
> > > $PRR = \frac{R_{random} - R_{uncertainty}}{R_{random} - R_{oracle}}$
> >
> > Given the definition in Malinin & Gales, $R_{uncertainty}$ is the area under the risk–coverage curve, this quantity is the same as AURC. Moreover, $R_{random}$ and $R_{oracle}$ depend only on model predictions and are independent of the uncertainty ranking. Under this formulation, PRR becomes an affine decreasing function of AURC for the same model predictions. Consequently, both metrics induce the same ordering of methods, and reporting one of them is sufficient.
> >
> > > ### ***3. Response to Weakness 3***
> >
> > Thank you for your valuable feedback. We agree that our original wording was too strong and we will revise it as follows:
> >
> > "Meanwhile, the strong correlation suggests that the performance of SemGrad is dependent on the semantic-preserving capability of the hidden states on which it operates, i.e., whether the hidden representations preserve semantic structure effectively. This observation is consistent with our core motivation that the output distribution of an LLM should be relatively stable under small semantic-preserving perturbations for confident inputs, rather than under arbitrary random perturbations."
> >
> > We hope this revision addresses the reviewer’s concern.
> >
> > > ### ***4. Response to Weakness 4***
> >
> > We thank the reviewer for this thoughtful observation. We agree that, in general, the norm of the parameter gradient does not admit a strict monotonic guarantee with respect to uncertainty. Importantly, parameter gradient methods (ExGrad) is used only as comparative baselines in our experiments, not as the conceptual foundation of our approach.
> >
> > In the work proposing ExGrad [2], which is formulated for classification tasks with a single ground-truth label, the authors derive an explicit decomposition of the parameter gradient with respect to the last-layer parameters (similar to LM head in the LLM case). This decomposition (eq. (5) in [2]) expresses the parameter gradient as the product of the last-layer hidden representation (corresponding to the hidden states before the LM head in the LLM setting) and a term that depends on the predictive distribution. This form reveals that, ***in general, sharper predictive distributions tend to produce gradients of smaller magnitude; however, this relationship is not guaranteed to be strictly monotonic.***
> >
> > When extending this perspective to language generation with large language models, the situation becomes substantially more complex due to the high-dimensional and sequential nature of the output space. Consequently, ***no general monotonicity guarantee can be expected for the parameter gradient in this setting.***
> >
> > Nevertheless, parameter gradient empirically demonstrates good performance in settings with a single ground-truth answer. Motivated by this observation, we introduce HybridGrad, which integrates the strengths of both parameter-gradient and our proposed SemGrad.
> >
> > [2] Igoe et al. (2022). How Useful are Gradients for OOD Detection Really?

---

> ### Author Response · Authors · 2025-11-21
> **Author Response III**
>
> > ### ***5. Response to Question 1***
>
> Thank you for your valuable suggestion. We provide the full results with ParaGrad included below and will add it to the main experiments.
> | Method     | Qwen3|  |  | Mistral|  |  | Llama3.1  |  |  |Avg.|
> |------------|----------------------|-------------|-------------|-----|-------------|-------------|---------|-------------|-------------|-|
> |            |SciQ | TriviaQA | TruthfulQA | SciQ | TriviaQ | TruthfulQA | SciQ | TriviaQA | TruthfulQA ||
> | SAR       | 72.72 | 81.52 | 67.98 |  76.57 | 85.23 | 68.55 |75.28 | 85.65| 64.44 |75.33|
> | ExGrad     | 71.34 | 80.37 | 63.77 |  77.53 | 84.53 | 66.40 |74.11 | 85.22 | 62.00 |73.92|
> | ParaGrad   | 72.09 | ***82.02*** | 66.40 |  ***77.99*** | ***85.91*** | 70.54 |74.98 | ***86.49*** | 63.91 |75.59|
> | SemGrad    | 72.20 | 80.40 | 69.06 |  75.55 | 82.37 | 72.27 |75.76 | 84.72 | 69.42 |75.75|
> | HybridGrad | ***72.83*** | 81.69 | ***69.61*** |  76.90 | 84.13 | ***72.72*** |***76.31*** | 85.89 | ***69.25*** |***76.59***|
>
> As shown above, ParaGrad performs well on datasets with a single ground-truth answer, especially on TriviaQA. On the contrary, SemGrad achieves stronger performance on datasets with high aleatoric nature (TruthfulQA). Combining them (HybridGrad) delivers a more stable and superior performance. These empirical results are fully aligned with our analysis in Section 3.3.
>
> > ### ***6. Response to Question 2***
>
> Yes, SemGrad Score and ParaGrad are not at the same level. Below is the average magnitude of SemGrad and ParaGrad of different models on TruthfulQA and TriviaQA:
> | Method     | Qwen3|   | Mistral|    | Llama3.1  |    |
> |------------|----|-------------|-----|------|---------|-------------|
> |            |TriviaQA | TruthfulQA |  TriviaQ | TruthfulQA |  TriviaQA | TruthfulQA |
> | ParaGrad   | 1.05e-5 | 6.40e-6 | 1.14e-5| 1.91e-5|6.77e-6|1.11e-5|
> | SemGrad    | 8.65e-4 | 6.21e-4 | 2.48e-3| 2.15e-3|5.92e-3|5.18e-3|
>
> The results show that SemGrad generally produces gradients several orders of magnitude larger than ParaGrad. As a consequence, SemGrad typically dominates the HybridGrad signal except in situations where the average per-token entropy-based gating strongly downweights it (e.g., under very sharp predictive distributions).
>
> This also reveals that the unstable behavior of SemGrad on the low aleatoric datasets might result from their numerical instability under extremely sharp predictive distributions. Thanks for your insightful observation and suggestion. We will add this discussion to the Appendix.
>
> ***Response to Question about Multiplicative Combination***
>
> Yes, a multiplicative combination can also be considered. In fact, a natural multiplicative variant is as follows:
>
> > $S_{para}^a \cdot S_{sem}^{(1-a)} = exp(a \cdot log(S_{para}) + (1-a)\cdot log(S_{sem}))$
>
> This shows that the multiplicative fusion is equivalent to an additive interpolation in log-space between the two scores. Given the small value of SemGrad and ParaGrad, log will compress the scale differences, which makes the multiplicative-based score more easily trusted in ParaGrad. We make a small experiment on SciQ and TruthfulQA for Llama3.1-8B-Instruct. Here is the result:
>
> | Method     | SciQ| TruthfulQA|
> |------------|----|-------------|
> | Addition-based   | 76.31|69.25|
> | Multiplicative-based | 76.01| 66.84|

---

> ### Author Response · Authors · 2025-11-21
> **Author Response VI**
>
> > ### ***7. Response to Question 3***
>
> Thank you for this thoughtful suggestion. In our current design, the interpolation weight is defined as
>
> > $\alpha = exp(-\bar{\omega})$
>
> where $\bar{\omega}$ is the average token entropy, used as a proxy for the sharpness of the sequence-level predictive distribution. The HybridGrad is then computed by:
> > $HybridGrad = \alpha * ParaGrad + (1-\alpha) * SemGrad$
>
>  This weighting is expected to behave robustly precisely because it depends on the relative sharpness of the predictive distribution, rather than on absolute token probabilities. Therefore, moderate implementation-level noise (e.g., differences in precision or minor numerical variations) does not lead to abrupt changes in the weighting, as long as the overall distributional shape remains stable.
>
>  Based on our discussion in "Response to  Question 2", if the output distribution is extremely sharp, $\bar{\omega}$ will be close to zero, and the weight will be close to 1, then HybridGrad relies primarily on ParaGrad. Conversely, when the distribution becomes more diffuse or multi-modal (high entropy), the method smoothly shifts towards SemGrad.
>
>  This is consistent with our motivation that ParaGrad will only be reasonable when there is only one ground truth answer, i.e., sharp predictive distribution.
>
> We agree with the reviewer that introducing an explicit temperature parameter,
> > $\alpha = exp(-\frac{\bar{\omega}}{\tau})$
>
> would make this trade-off more explicit and would allow systematic control. Smaller values of $\tau$ bias the estimator towards SemGrad, while larger values place more weight on ParaGrad.
>
> However, while such a temperature can offer finer control, it also introduces an additional hyperparameter. For the current version, we intentionally avoid this extra degree of freedom in order to preserve the hyperparameter-free nature and cross-dataset robustness of the method. Given the consistent empirical performance observed across models and tasks, we believe the fixed form provides a reasonable balance between flexibility and simplicity.
>
> Thanks again for your valuable and insightful suggestion.
>
> > ### ***8. Response to Question 4***
>
> Thank you for this important question. We do observe a dependency on response length for gradient-based estimators since gradient signals can be diluted across numerous correct and less informative tokens. Our introduced token weight helps, but the performance still drops for long generations. That is why we claim in the limitation that "our work primarily targets claim-level predictions (i.e., short answers)" (line 679). However, this does not diminish its practical usefulness, since claim-level evaluation is widely used as a fundamental building block for long-form UQ methods (e.g., as practiced in the “Detecting Confabulations in Biographies” section of [3]). The efficiency of our proposed gradient-based method makes it a practical and competitive solution in real-world settings.
>
> > ### ***9. Response to Question 5***
>
> No, we do not leverage the negative answers. We compute the BEM score between the model-generated answer and each reference correct answer provided by TruthfulQA, and report the maximum score across references.
>
> > ### ***10. Response to Question 6***
>
> Thank you for your valuable feedback. We prompt the DeepSeek API with a few-shot examples to generate five paraphrases for each query, as described in Appendix C.1.
>
> We conduct a small-scale validation on TruthfulQA to assess how well the generated paraphrases preserve semantic meaning. We evaluate semantic consistency using two independent methods: (i) an NLI-based judge (DeBERTa-large trained on MNLI), where we assign a score of 1 if the paraphrase is classified as entailment, and 0 otherwise; and (ii) an LLM-based judge (Llama3-Instruct-70B), where we prompt the model with a Yes/No question regarding semantic equivalence, assigning 1 if the response contains “Yes”.
>
> The results are shown below:
>
> | NLI-as-judge| LLM-as-judge|
> |------------|------------|
> | 90.08| 98.72|
>
> These results indicate that the DeepSeek-generated paraphrases largely preserve the original semantics. We will add this experiment to the appendix.
>
>
> [3] Farquhar et al. (2024). Detecting hallucinations in large language models using semantic entropy. Nature 2024.

---

> ### Author Response · Authors · 2025-11-21
> **Author Response V**
>
> > ### ***11. Response to Question 7***
>
> Thank you for the valuable feedback. Following the same token-weighting scheme used in SAR, we replace the entropy-based weights with the semantic-importance weights and evaluate the resulting variants (denoted SemGrad-SAR and HybridGrad-SAR) on TruthfulQA using Llama3.1-8B-Instruct.
> | UQ Method  | TruthfulQ  |
> |------------|------------|
> | SAR    |  64.44 |
> | ExGrad    | 62.00 |
> | SemGrad  |  69.42 |
> | SemGrad-sar|69.02|
> | HybridGrad |69.25 |
> | HybridGrad-sar |**69.45**|
>
> We observe that SAR-style weighting yields only marginal differences compared to our entropy-based weighting.
>
> > ### ***12. Response to Question 8***
>
> We follow the core code of this repository "https://github.com/jlko/semantic_uncertainty" which is the nature version.
>
> > ### ***13. Response to Question 9***
>
> Thank you for the valuable suggestion. In our experiments, we only report length-normalized predictive entropy (LN-PE), as length normalization is a standard and widely adopted practice for sequence-level uncertainty estimation, and LN-PE generally performs better than PE.
>
> For completeness, we additionally evaluated PE without length normalization. The results are shown below:
>
> | UQ Methods | Qwen3| |  | Mistral | |  | Llama3.1 |  |  | Avg. |
> |-----------|-----------------------|---------------------------|-----------------------------|------------------------|---------------------------|-----------------------------|-------------------------------|---------------------------------|-----------------------------------|------|
> |      | SciQ | TriviaQ | TruthfulQ | SciQ | TriviaQ | TruthfulQ| SciQ | TriviaQ | TruthfulQ||
> | LN-PE     | 67.08 | 80.00 | 64.78 | 76.68 | 84.02 | 66.29 | 72.51 | 84.53 | 63.38 | 73.25 |
> | PE| 67.42 | 78.81| 61.19| 76.26| 83.46|63.82|73.37|84.58|61.05|72.22|
> | SemGrad   | 72.20 | 80.40 | 69.06 | 75.55 | 82.37 | 72.27 | 75.76 | 84.72 | 69.42 | 75.75 |
> | HybridGrad | 72.83 | 81.69 | 69.61 | 76.90 | 84.13 | 72.72 | 76.31 | 85.89 | 69.25 | 76.59 |
>
> These results confirm that length normalization improves PE.
>
> Regarding the suggested LN-SE, I am a bit confused since Semantic Entropy already normalizes the generation logprob before computing the entropy, as shown in https://github.com/jlko/semantic_uncertainty/blob/master/semantic_uncertainty/compute_uncertainty_measures.py line 243. Could you clarify what specific form of LN-SE is desired?
>
> > ### ***14. Response to the Notes in Weakness***
>
> Thank you very much for your careful review and helpful suggestions.
> - We will replace all equation references with \eqref to ensure consistent LaTeX formatting.
> - We agree that the current colormap may suggest an unintended correlation structure. In the revision, we will switch to a colormap with a single color for Figures 2–4 to avoid misinterpretation.
> - Thank you for your suggestion. We will add a discussion in the appendix comparing our findings to their results and clarifying both the similarities and differences in terms of layer-wise semantic representations.
> - Thank you for your feedback. We will correct the phrase to “semantic-preserving perturbations.”
>
>
> ### ***We hope that our clarifications resolve the reviewer’s concerns. If any aspects of our explanation remain unclear, or if there are further concerns, we would be happy to address them or provide additional experiments. We thank the reviewer again for their time and constructive feedback.***

---

> > ### Comment · Reviewer_QuTT · 2025-11-24
> >
> > Thank you for your detailed feedback to my review. If not stated otherwise, the provided response has alleviated my concerns.
> >
> > 5 - Thank you, as I expected a lot of performance on SciQA and TriviaQA comes from the ParamGrad component. I would urge the authors to present SemGrad more in light of being a way to overcome the limitations of ParamGrad in the setting of multiple ground-truth answers, combining the two to HybridGrad as both have their strengths and weaknesses. I would also like to note that SemGrad alone is never best on any of the considered model/task combinations, which would warrant a more detailed analysis of why HybridGrad surpasses the individual scores its comprised of, e.g. through an ablation with the suggested temperature parameter (what is discussed in your answer 7).
> >
> > 6 - Very interesting insight. Would you mind providing the average per token entropy as well to get a feeling of what of the two will dominate in practice? Even better would be a plot with histograms over the average per token entropy.
> >
> > 7 - While I agree that the finally evaluated method should stay hyperparameter-free by stating $\tau=1$, I would still find it valuable for the ablation stated in my answer to 5).
> >
> > 11 - Would you mind providing results for ParamGrad and ParamGrad-sar as well?
> >
> > 13 - I see, if the reported SE results use length-normalization already, I would suggest to also show results without length-normalization. While length-normalization improved PE on average in your reported results, it is sometimes worse on individual task/model combinations. In general, results on whether length-normalization helps or not have been inconclusive in previous studies. As this is relatively low effort, I was interested in its effect under your experimental setup.
> >
> > Again, thank you for providing detailed answers to my questions. I will raise my score to 6 to reflect my updated assessment of the paper.

---

> ### Author Response · Authors · 2025-12-03
> **Author Response VI**
>
> Thank you very much for raising the score. Here are our responses to the reviewer's additional questions:
>
> - Response to 5. We agree that SemGrad and ParaGrad have complementary strengths, as discussed in Section 3.3. SemGrad is theoretically well-motivated in both low- and high-aleatoric settings, but its empirical performance is less stable than ParaGrad in low-aleatoric settings, where ParaGrad is also theoretically justified. When the aleatoric nature of the data to be evaulated is known in advance, one can choose the more suitable estimator accordingly. When this is unknown, HybridGrad combines the strengths of both and yields a more robust metric that generalizes better across datasets. We add a paragraph in Section 3.3 (lines 301–309) that explicitly analyzes the strengths and weaknesses of the parameter-gradient method and SemGrad, and explains our motivation for combining them.
> - Response to 6 and 7. We thank the reviewer for this valuable suggestion. We now plot histograms of the average per-token entropy $\bar{\omega}$ in Figure 5 and additionally analyze the influence of the balancing weight $e^{-\bar{\omega}}$ by introducing additional hyperparameters. The detailed analysis and ablations are provided in Appendix D.3 of the revised version
>
> - Response to 11. Here is the result of ParaGrad and ParaGrad-sar. We still observe a marginal difference.
>
>
>   | UQ Method  | TruthfulQ  |
>   |------------|------------|
>   | SAR    |  64.44 |
>   | ExGrad    | 62.00 |
>   | SemGrad  |  69.42 |
>   | SemGrad-sar|69.02|
>   | ParaGrad | 63.91|
>   | ParaGrad-sar| 64.70|
>   | HybridGrad |69.25 |
>   | HybridGrad-sar |**69.45**|
>
> - Response to 13. Here is the result of Semantic Entropy without length normalization. Similar to PE, length-normalization improved performance on average, but sometimes performance was worse on individual task/model combinations.
>
>   | UQ Methods | Qwen3| |  | Mistral | |  | Llama3.1 |  |  | Avg. |
>   |-----------|-----------------------|---------------------------|-----------------------------|------------------------|---------------------------|-----------------------------|-------------------------------|---------------------------------|-----------------------------------|------|
>   |      | SciQ | TriviaQ | TruthfulQ | SciQ | TriviaQ | TruthfulQ| SciQ | TriviaQ | TruthfulQ||
>   | S.E.| 56.88 | 76.16| 63.10| 68.53| 80.64 |66.71|70.27|83.12|59.59|69.45|
>   | S.E. w/o LN| 55.61 | 75.79| 61.24| 69.15| 79.82 |66.12|71.72|83.38|57.28|68.90|

---

> ### Author Response · Authors · 2025-12-03
> **Summary of Changes in the Revised Submission**
>
> We have revised the original submission based on Reviewer QuTT’s valuable feedback and our responses. Specifically:
> - We add experiments using LLM-as-a-judge for correctness evaluation to Appendix D.1.
> - We include additional experimental results using complementary evaluation metrics (AURC, AUCPR) in Appendix D.2.
> - We revised the description in Section 4.3 (lines 417-421) following our response to weakness 3.
> - We add ParaGrad to the main experiments. Specifically, we explicitly introduce ParaGrad and provide the formula in Section 3.3 (lines 312-322). We update Table 1 and modify the analysis in Section 4.2.
> - We replace all equation references with \eqref to ensure consistent LaTeX formatting.
> - We switch to a colormap with a single color for Figures 2–4 to avoid misinterpretation.
> - We add a paragraph in Related Work to discuss the similarities and differences with INSIDE.
> - We add a section in Appendix D.3 to additionally analyze the influence of the balancing weight $e^{-\bar{\omega}}$ by introducing additional hyperparameters.
> - We add a paragraph in Section 3.3 (lines 301–309) that explicitly analyzes the strengths and weaknesses of the parameter-gradient method and SemGrad, and explains our motivation for combining them.
> - We add the experiment of validating the quality of our generated paraphrases (response to Question 6) to Appendix C.1 (lines 784-790).
>
> We sincerely thank Reviewer QuTT for their constructive feedback.

---

### Official Review · Reviewer_hgm6 · 2025-10-30

**Soundness:** 3
**Presentation:** 2
**Contribution:** 2
**Rating:** 4
**Confidence:** 4

**Summary:**

The paper proposes gradient based uncertainty estimation for natural language generation problems.
They first introduce the Semantic Preservation Score (SPS), which is later used to determine the parts of the gradient of the model/intermediate states most correlated with semantics.
Then two gradient based uncertainty estimation methods are introduced: SemGrad and HybridGrad.
The former uses the gradient with respect to hidden states and the latter additionally incorporates model weight gradients, which was already done in prior work.
The authors perform evaluation of the proposed approach to verify its utility.

**Strengths:**

This work explores uncertainty quantification by gradient, the role of model parameter and hidden state gradients.
The concept of using the KV cache gradients is generally appealing and interesting, although a lot of heuristics are involved.
Many good recent method for UE are compared to in the evaluation.
Authors acknowledge the limitation of applicability of their method to only the white-box scenario and perform several ablations.

**Weaknesses:**

1. The term "semantic" is mentioned 112 times in the paper. "Semantics" are used in arguments and even backpropagated through. I believe a paragraph of discussion of what is "semantic" deserves to be in the introduction or preliminaries.
2. The SPS motivation (i.e. why should we do it this way, why not some other way, i.e. through an embedding model or so) does not feel well conducted.
3. Figure 1: Technically and lexically awkward labels: "Certain input", "Uncertain input". In line 144 it is the same: "For a certain input x" - does it now mean that we have a fixed input x or that the input x is supposed to be "certain".
4. Experimental validation.
    1. Experimental suite is narrow (3 models / 3 datasets, all short answer QA), for a method that is heavily heuristic driven, the breadth of evaluation is important.
    2. The selective prediction experiments hinge on a single, not very commonly used correctness function - BEM. This is prone to skew the results on QA datasets (see [1][2]).
    3. Line 316: "Many of the questions in TruthfulQA are open-ended (e.g., “What happens to you if you eat watermelon seeds?”), which naturally introduces a high degree of aleatoric uncertainty." TruthfulQA correctness has little to do with aleatoric risk (i.e. the risk of model just decoding the wrong thing) and more with training data selection / order during pretraining. I view connecting it to aleatoric risk as a bad practice.

### References
1. Ielanskyi, M., Schweighofer, K., Aichberger, L. & Hochreiter, S. Addressing Pitfalls in the Evaluation of Uncertainty Estimation Methods for Natural Language Generation. Preprint at https://doi.org/10.48550/arXiv.2510.02279 (2025).
2. Santilli, A. et al. Revisiting uncertainty quantification evaluation in language models: Spurious interactions with response length bias results. in Proceedings of the 63rd annual meeting of the association for computational linguistics (volume 2: Short papers) (eds Che, W., Nabende, J., Shutova, E. & Pilehvar, M. T.) 743–759 (Association for Computational Linguistics, Vienna, Austria, 2025). doi:10.18653/v1/2025.acl-short.60.

**Questions:**

1. Could you provide a more detailed explanation on the type of uncertainty in TruthfulQA? How would risk of decoding one option from the training set vs another option be correlated to the aleatoric risk?
2. What motivated the choice of BEM as a correctness metric for selective prediction experiments? Would results look different with e.g. AlignScore or LLM as a Judge?
3. Why does the method appear to underperform on Mistral-Nemo-Instruct 12B?
4. How does the methods compare in terms of compute and, more importantly, memory requirements compared to prior works?
5. Fig. 2 shows what I would view as a a sink token effect - the 'technical' tokens eat up a lot of attention mass. How is this accounted for in SemGrad?

---

> ### Author Response · Authors · 2025-11-21
> **Author Response I - Clarification on Definition**
>
> Thank you for your valuable feedback and constructive suggestions. We appreciate the reviewer’s careful evaluation. Below is our response to your comments, which we hope will answer your questions and concerns.
>
> > ### ***1. Response to the Definition of “Semantics” (Weakness 1)***
>
> Thank you for pointing this out. We agree that the notion of ‘semantic’ plays a central role in our method and appears frequently throughout the paper. In this work, we use ‘semantic’ in an operational sense: ***it refers to the underlying meaning conveyed by the text, distinct from surface-form or syntactic variation***. Accordingly, when we refer to semantically equivalent input perturbations, we mean inputs that express the same meaning despite differences in wording.
>
> Since computing gradients directly over linguistic meaning is infeasible, our method relies on a numerical proxy for semantic information within the LLM’s computation. We use hidden representations that remain stable under meaning-preserving perturbations as semantic-preserving embeddings: representations where semantically similar inputs are close, and semantically dissimilar ones are separated under a similarity metric. This proxy allows us to operationalize semantic reasoning in a gradient-based framework.
>
> In the revision, we will modify the introduction and Section 3, clarifying this operational definition of ‘semantic’ and how it is instantiated through the model’s hidden states.
>
> > ### ***2. Response to Concerns on TruthfulQA (Question 1 and Weakness 4.1)***
>
> Thank you for your valuable feedback. We fully agree that the origin of errors in TruthfulQA can be multifaceted and may involve memorization effects, training data selection, or optimization dynamics. However, our use of ***“aleatoric uncertainty” does not describe the cause of errors in the learned model, but rather a structural property of the underlying language data.***
>
> Aleatoric uncertainty in our work follows the standard definition [1]: ***the inherent, irreducible randomness within the data***. In the context of language, it corresponds to the inherent one-to-many mapping, where a single input query can admit multiple plausible or equally valid outputs. When we refer to a query or dataset as having high aleatoric uncertainty, we mean that the underlying ground-truth language generation distribution $p^*(y|x)$ is intrinsically multi-modal, rather than concentrated around a single canonical answer. Therefore, ***aleatoric uncertainty in our work refers to the property of the underlying ground-truth language generation distribution (the language data) rather than the model we have learned.***
>
> This distinction explains why parameter gradient is misleading in high-aleatoric settings such as TruthfulQA. The gradient of the output log-probability with respect to model parameters measures how much the probability of a particular sampled sequence $y$ can be increased. As long as this probability is not exactly one, the gradient remains non-zero. Therefore, parameter gradients actually reflect the sharpness of the predictive distribution, rather than the discrepancy between the learned model and the ground-truth distribution.
>
> However, in high-aleatoric regimes, the ground-truth distribution is inherently non-sharp and multi-modal. Even when the learned model closely matches this ground-truth distribution, the parameter gradients can remain large, leading to biased uncertainty. In contrast, SemGrad does not depend on assumptions of output sharpness or unimodality, and is therefore more robust in high-aleatoric settings. This behavior is consistently supported by our empirical results, where SemGrad substantially outperforms parameter-gradient baselines on TruthfulQA.
>
> Thanks again for your feedback, we will revise the introduction part to clarify the definition of the aleatoric uncertainty.
>
> [1] Hullermeier & Waegeman (2021). Aleatoric and epistemic uncertainty in machine learning: an introduction to concepts and methods. Machine Learning.

---

> ### Author Response · Authors · 2025-11-21
> **Author Response II - Response to Concerns on Semantic Preservation Score (SPS)**
>
> > ### ***3. Response to SPS motivation (Weakness 2)***
>
> Thank you for the valuable feedback. Our goal in this work is not simply to measure semantic similarity, but to find a ***differentiable*** semantic space inside the LLM that supports gradient-based uncertainty estimation.
>
> As described in Section 3.1, the semantic-preserving embedding $h_{E}(x)$ must satisfy two requirements:
> 1. It must be produced by the model’s own forward computation, so that gradients with respect to this representation are well-defined and directly connected to the model’s prediction behavior.
> 2. It must exhibit semantic consistency, meaning that ***semantically equivalent inputs are mapped to nearby representations, while semantically dissimilar inputs are mapped to distant ones*** (Section 3.1, line 181). This property allows gradients taken in this space to be interpreted as sensitivity to small semantic-preserving interventions.
>
> These requirements make off-the-shelf embedding models (e.g., SBERT, SimCSE, CLIP-style encoders) unsuitable in our setting: although they measure semantic similarity, they are not part of the forward graph of the target LLM, and thus cannot be used to compute meaningful gradients of the model’s predictive distribution.
>
> SPS is introduced as a principled criterion to identify such an internal representation that ***maps semantically equivalent variants of input closer and maps semantically different inputs distant.*** For each candidate hidden state, we measure:
>
> - the average distance between ***semantically equivalent*** variants ($S_{within}$), and
>
> - the average distance between ***semantically inequivalent*** inputs ($S_{across}$)
>
> The difference between these quantities directly reflects the ***extent to which the representation preserves semantic structure***. We therefore select the hidden state with the largest separation, yielding an embedding space where small perturbations can be interpreted as semantic-preserving interventions.
>
> Finally, our ablation results (Section 4.3 and Figure 3) show a strong correlation between SPS scores and SemGrad performance, providing empirical support for both our design choice and its underlying motivation.
>
> > ### ***4. Response to Concerns About Sink Token Effects for SPS score in Figure 2 (Question 5)***
>
> Thank you for your insightful feedback. As stated above, SPS measures the extent to which the representation preserves semantic structure, i.e., measures the semantic discriminability of a hidden representation by comparing:
>
> - the average distance between semantically equivalent variants ($S_{within}$), and
>
> - the average distance between semantically inequivalent inputs ($S_{across}$)
>
> A larger value of SPS (corresponding to the red section in Figure 2) indicates better discriminability.
>
> If a given token or hidden state were merely acting as an “attention sink” without encoding semantic information, it would absorb attention equally for both semantically similar and semantically different inputs. In that case, the difference $S_{within} - S_{across}$ would remain close to zero, and SPS would not be large.
>
> However, we observe consistently high SPS values for specific hidden states, indicating that these representations systematically map semantically equivalent variants of input closer and map semantically different inputs distant. This suggests that ***SPS is capturing meaningful semantic structure rather than purely structural attention artifacts.***
>
> Finally, our ablation results (Section 4.3 and Figure 3) show a strong correlation between SPS scores and SemGrad performance, providing empirical support that SPS identifies semantically informative representations rather than sink-token artifacts.

---

> ### Author Response · Authors · 2025-11-21
> **Author Response III - Response to Evaluation Robustness**
>
> > ### ***5. Response to Concerns on Correctness Metric BEM (Weakness 4.3 and Question 2)***
>
> Thank you for your valuable feedback. We fully agree that correctness metrics can significantly affect selective prediction results. Lexical–based metrics, which are commonly used in our baselines [2–6], are known to be unreliable in open-ended QA.
>
> On the contrary, BEM has been shown to be a reliable semantic matching metric for evaluating the correctness of short-form QA answers. Specifically, we followed prior work by Kamalloo et al. (2023)[7] (290 citations), which systematically compared automatic metrics (including LLM-as-judge) against human annotations, and here is their conclusion in the introduction:
>
> > "automated evaluation mechanisms such as BEM based on semantic matching between the gold answers and generated answers produce a relative performance that is mostly consistent with human evaluation."
>
> This conclusion does not hold for long-form generation. However, our work focuses on claim-level, short-answer settings (as discussed in the Limitations section) and, therefore, aligns with the above conclusion.
>
> We agree that LLM-as-a-judge can provide more fine-grained judgments; however, BEM is ***computationally lightweight, reproducible, and cost-free***, and prior work has shown it to be an effective metric for evaluating short-form QA answers, which motivated our choice.
>
> To address your concern, we conducted additional experiments using an LLM-as-a-judge evaluator (with DeepSeek API) on the same generations from Llama3.1-8B-Instruct. The results, compared with BEM-based evaluation, are shown below:
>
> | UQ Method   | SciQ & LLM | SciQ & BEM | TriviaQ & LLM | TriviaQ & BEM | TruthfulQ & LLM | TruthfulQ & BEM | LLM Avg. | BEM Avg. |
> |------------|------------|------------|---------------|---------------|----------------|----------------|------|--|
> | LN-PE      | 73.23 | 72.51 | 86.10 | 84.53 | 57.26 | 63.38 |  72.20 | 73.47|
> | S.E.       | 74.06 | 70.27 | 85.92 | 83.12 | 58.33 | 59.59 |  72.77 | 70.99|
> | S.D.       | 75.73 | 74.00 | 84.39 | 82.44 | 53.59 | 57.75 | 71.24 | 71.40|
> | M.I.       | 73.57 | 72.43 | 84.60 | 83.52 | 55.01 | 64.25 | 71.06 | 73.40|
> | **SAR**    | 76.76 | 75.28 | 86.82 | 85.65 | 59.07 | 64.44 | 74.22 | 75.12|
> | ExGrad     | 73.87 | 74.11 | 86.35 | 85.22 | 57.27 | 62.00 | 72.50| 73.78|
> | **SemGrad** | 77.42 | 75.76 | 85.76 | 84.72 | **65.97** | **69.42** | 76.38| 76.63|
> | **HybridGrad** | **77.76** | **76.31** | **87.03** | **85.89** | 65.35 | 69.25 | ***76.71***| ***77.15***|
>
> As shown, the rankings and relative trends under BEM and LLM-as-a-judge are highly consistent, and our proposed methods continue to achieve superior performance under both metrics. This suggests that our main conclusions are robust to the choice of correctness metric.
>
> > ### ***6. Response to Breadth of Evaluation (Weakness 4.2)***
>
> Thank you for your valuable feedback. We agree that a broader experimental suite is generally desirable. Nevertheless, we believe our current experiments are representative to validate both the motivation and effectiveness of our method for the following reasons:
>
> - We evaluate our approach across three different LLMs with varying settings.
>
> - We conduct experiments on widely used, representative datasets that include both low-aleatoric (SciQ, TriviaQA) and high-aleatoric (TruthfulQA) datasets.
>
> We acknowledge that our experiments focus on short-answer QA tasks. As discussed in the Limitations section, our method is specifically designed for claim-level (short-form) outputs rather than long-form generation. However, this ***does not diminish its practical usefulness***, since claim-level evaluation is widely used as a fundamental building block for long-form UQ methods (e.g., as practiced in the “Detecting Confabulations in Biographies” section of [8]). The efficiency of our proposed gradient-based method makes it a practical and competitive solution in real-world settings.
>
> [2] Duan et al. (2024). Shifting attention to relevance: Towards the predictive uncertainty quantification of free-form large language models. ACL 2024.
>
> [3] Abbasi-Yadkori et al. (2024). To believe or not to believe your LLM: iterative prompting for estimating epistemic uncertainty. NeurIPS 2024.
>
> [4] Kuhn et al. (2023). Semantic uncertainty: linguistic invariances for uncertainty estimation in natural language generation. ICLR .
>
> [5] Chen et al. (2024). INSIDE: llms’ internal states retain the power of hallucination detection. ICLR 2024.
>
> [6] Qiu et al. (2024). Semantic density: Uncertainty quantification for large language models through confidence measurement in semantic space. NeurIPS 2024.
>
> [7] Kamalloo et al. (2023) Evaluating open-domain question answering in the era of large language models. ACL 2023.
>
> [8] Farquhar et al. (2024) Detecting hallucinations in large language models using semantic entropy. Nature 2024.

---

> ### Author Response · Authors · 2025-11-21
> **Author Response IV - Response to Other Concerns**
>
> > ### ***7. Response to Computation Efficiency and Memory Efficiency (Question 4)***
>
> - ***About Computation Efficiency***
>
>   We report empirical runtime comparisons in Appendix B. Both proposed gradient-based methods (SemGrad and HybridGrad) consistently run faster than sampling-based baselines.
>
>   Our current implementation relies on PyTorch’s automatic differentiation and does not explicitly specialize gradient computation to a single token position. As a result, the reported runtimes can be viewed as a conservative estimate of the method’s efficiency. Despite this, SemGrad still outperforms sampling-based baselines, which indicates the computational efficiency of our proposed method.
>
>  - ***About Memory Efficiency***
>
>    Our method requires a single forward and backward pass through the model, which does incur additional memory overhead for storing activations, similar to a standard training step. Concretely, the memory scales as $O(L \cdot T \cdot D)$, where L is the number of layers, T the sequence length, and D the hidden size. In principle, the dependence on T can be further reduced since gradients are only required at a small number of token positions.
>
>    In contrast, while sampling-based methods do not require storing backward activations, they require K independent forward passes with K generated outputs; memory scales as $K \cdot O(L \cdot T \cdot D)$. Many methods additionally store per-sample embeddings or similarity structures and, in some cases, rely on auxiliary models for semantic comparison. As a result, their memory grows with the number of samples K, and in many cases, includes additional storage for other operations.
>
>   Therefore, our method remains more memory- and compute-efficient than sampling-based approaches that scale with the number of samples. Thank you for your suggestion, a discussion of memory efficiency will be added to the Appendix.
>
> > ### ***8. Response to Question 3***
>
> We can only provide a hypothesis. Given the complex structure of LLMs, the result can be a consequence of some intrinsic preference that can not be explicitly explained.
>
> Here is a plausible explanation. As you can see in Figure 2, both Llama3.1 and Qwen3 encode semantics (based on SPS score) with an additional special token (both of them are the start token of assistance). In contrast, Mistral-Nemo-Instruc does not have such a kind of special token, and the best semantic-preserving token is the last user input token, which is different for each input. Therefore, unstable input tokens at this position may lead to unstable numerical behavior, i.e., the hidden states might encode noise information other than the semantics, and can explain the observed performance gap.
>
> Despite this, our method remains competitive and outperforms most baselines on Mistral, while still achieving a substantial margin in high-aleatoric settings.
>
> > ### ***9. Response to Weakness 3***
>
> Thank you for your valuable feedback and for pointing out this ambiguity. In Figure 1, “Certain input” was intended to refer to an input on which the model is confident, and “Uncertain input” to an input on which the model is uncertain; these labels were abbreviated for brevity in the figure. In contrast, in line 144, the phrase "For a certain input $x$" was intended to mean “for a given input
> x”, i.e., a fixed input, rather than implying model confidence.
>
> We are sorry for the confusion and will revise the figure labels and the text in line 144 to remove this ambiguity. Thank you for highlighting this issue.
>
> ### ***We hope that our clarifications resolve the reviewer’s concerns. If any aspects of our explanation remain unclear, or if there are further concerns, we would be happy to address them or provide additional experiments. We thank the reviewer again for their time and constructive feedback.***

---

> ### Author Response · Authors · 2025-12-03
> **Summary of Changes in the Revised Submission**
>
> We have revised the original submission based on Reviewer hgm6’s valuable feedback and our responses. Specifically:
>
> - We bold the key sentence stating our basic assumption in Section 3.1 and emphasize the equivalence between “semantics” and “underlying meaning” (lines 170–172).
>
> - We modify the caption of Figure 1 and change “For a certain input” to “For an input that the model is certain about” (line 71). We also change “For a certain input” in Section 2 to “For a specific input” (line 144).
>
> - We add a clear definition of aleatoric uncertainty as “inherent, irreducible randomness within the data,” together with a reference, in the introduction (line 45).
>
> - We add experiments using LLM-as-a-judge for correctness evaluation to Appendix D.1.
>
> - We add our analysis of memory efficiency (Response to Question 4) to Appendix B and restructured this appendix accordingly.
>
> We sincerely thank Reviewer hgm6 for their constructive feedback.

---

### Official Review · Reviewer_C28m · 2025-10-30

**Soundness:** 3
**Presentation:** 4
**Contribution:** 3
**Rating:** 4
**Confidence:** 4

**Summary:**

The authors propose SemGrad, a gradient-based UQ method that operates in semantic space to estimate uncertainty for NLG tasks. The method first identifies the highest semantic preserving token of an input (expected highest value of the proposed metric SPS), whose embeddings best represents the entire input’s semantics. Then it computes the gradient of the log-likelihood of the output sequence with respect to the top half layer embeddings of the identified semantic preserving token. This measures how sensitively the model’s output sequence changes when perturbing the input while preserving its semantics.
Additionally, the authors also propose HybridGrad, a combination of SemGrad and parameter gradients.

**Strengths:**

- The paper is very well written and has a clear structure
- The motivation for SemGrad is clear and neat (sections 3.1 and 3.2 are well preparing for the methods)
- The ablations are insightful

**Weaknesses:**

**Method**:
- The paper’s central argument is that NLG tasks require a fundamentally different treatment from classification tasks. The authors claim that using parameter gradients as a proxy for uncertainty “does not extend to the ground-truth distribution of natural language” (line 156) and that “the parameter gradient norm can be misleading” (line 159). The main premise is therefore to "overcome the limitations of the parameter gradient" (line 164), with substantial effort devoted to motivating, deriving, and analyzing the Semantic Preservation Score (SPS) and the resulting SemGrad method. My main concern is that the authors subsequently reintroduce parameter gradients in the HybridGrad variant, presented only briefly at the end of the methodology section. The justification for this addition is unconvincing and lacks theoretical grounding. It is unclear why, from a conceptual standpoint, switching to parameter gradients when the model’s predictive distribution is already sharp and stable should provide any benefit over SemGrad alone. While HybridGrad empirically improves performance, this improvement contrasts with the earlier theoretical motivation and raises questions about whether the empirical evaluation fully isolates the intended effect.
- Another technical concern is that the entropy weights $w_t$ depend on the same predictive distribution as the log-likelihood terms. Unless the authors explicitly detach these values from the computation graph, the weighting would alter the gradient itself. On the other hand, if the entropy weights are detached, it would ignore the entropy’s own sensitivity to the embeddings.


**Evaluations**: The baselines should be extended by UQ methods that also do not require sampling multiple output sequences (e.g., G-NLL [1]).

---
[1] Lukas Aichberger, Kajetan Schweighofer, and Sepp Hochreiter. Rethinking uncertainty estimation in natural language generation. arXiv preprint arXiv:2412.15176, 2024.

**Questions:**

- Why does the "intuition that uninformative tokens (e.g., stopwords or subwords) always exhibit low output entropy" (line 282) motivate the use of the above mentioned entropy weights? On the one hand, in the sentence “I think it’s going to rain,” the token “think” carries little factual meaning. However, a model might assign high entropy since many similar verbs could fit (“guess”, “believe”, “assume”, “suppose”). On the other hand, in the sentence “The capital of Australia is Sydney,” the token “Sydney” might have low entropy if the model is confidently wrong.

---

> ### Author Response · Authors · 2025-11-21
> **Author Response I - Response to Weakness of Method 1**
>
> Thank you for your thoughtful feedback and constructive suggestions. We appreciate the reviewer’s careful evaluation. Below is our response to your comments, which we hope will answer your questions and concerns.
>
> > ### ***1. Response to Weakness of Method 1***
>
> Thank you for your insightful feedback. Our argument is NOT that parameter gradients are universally invalid for all NLG tasks, but that they become unreliable ***specifically in the presence of high aleatoric uncertainty***, where multiple semantically valid answers exist (section 2, lines 156-161). Under the low aleatoric setting (e.g., QA task with one ground-truth answer), parameter gradient remains a valid measure.
>
> This claim is empirically supported by our comparison across two types of datasets:
> - Single ground-truth answer datasets (SciQ, TriviaQA)
> - Multiple plausible answer datasets (TruthfulQA), which exhibit high aleatoric uncertainty.
>
> Our results show that:
> - Parameter-gradient method (ExGrad) indeed performs well in the single ground-truth setting, where SemGrad also achieves comparable performance.
> - In high-aleatoric settings, SemGrad remains stable while Parameter gradient does not (SemGrad outperforms Parameter gradient by a large margin).
>
> This aligns with our theoretical motivation that “semantic gradients remain valid even under high aleatoric uncertainty” (line 195).
>
> Notably, although SemGrad is in principle valid in the single ground-truth answer setting, we found that its empirical performance is sometimes inferior to ParaGrad. We hypothesize the reasons are:
>
> - Under the single ground-truth answer case, Parameter gradient aligns directly with the model’s training objective and thus can provide greater numerical stability (as stated in line 294).
> - SPS identifies the hidden states that best capture semantics, but it does not guarantee that the hidden state only captures semantics, which may introduce empirical instability.
>
> Therefore, we interpolate between ParaGrad and SemGrad based on sequence-level entropy. The underlying rationale is as follows:
> - In the single ground truth answer case (recognized by low sequence-level entropy), both estimators are valid, but ParaGrad is more stable, so we up-weight ParaGrad.
> - In the multiple correct answer case (recognized by high sequence-level entropy), ParaGrad becomes ill-defined and SemGrad provides better theoretical guarantees, so we rely more on SemGrad.
>
> The empirical improvements are consistent with this design.
>
> While the current implementation of SemGrad is not a perfect solution, it introduces a promising and principled direction: viewing LLM uncertainty through the lens of semantic gradients. Our experiments and ablations consistently support the effectiveness of this perspective.

---

> ### Author Response · Authors · 2025-11-21
> **Author Response II - Response to Weakness of Method 2**
>
> > ### ***2. Response to Weakness of Method 2 and Questions about the Entropy Weight***
>
> Thank you for your valuable observation and feedback.
>
> - About gradient flow:
>   We indeed freeze the entropy weight (detach them from the computation graph). You can find this operation in our provided code (./uncertainty/uncertainty_estimation/gradient/utils.py line 146). Therefore, they serve purely as constant token weights and do not alter the gradient structure. We will revise the description in section 3.3.
>
> - Why entropy weighting is used:
>
>    The purpose of the entropy weight is to downweight ***extremely uninformative tokens***, such as articles (“the”, “a”), punctuation (",", "." ), prepositions ("to", "by"), or some filler subwords (“-ing”). These tokens often exhibit near-zero entropy because the model assigns nearly all the probability mass to a single token. For example, in the response "Nervous tissue." generated by Llama3.1-8b-instruct:
>
>   | Token| Entropy| Probability|
>   | -- | -- | -- |
>   |N| 0.8780| 0.7845|
>   |erv|0.2287|0.9398|
>   |ous|0.0108|0.9990|
>   |Ġtissue|0.2672|0.9458|
>   |.|0.0383|0.9939|
>
>    tokens such as punctuation (. with entropy 0.0383) and filler subwords (ous with entropy 0.0108) are indeed near-zero. These are the primary cases where entropy weighting is beneficial.
>
>    Importantly, token entropy does not collapse to zero unless the distribution is extremely sharp (e.g., >99% mass on one token).  For instance, even a distribution that places 95% of its mass on one token still yields an entropy of approximately 0.2, as shown in the table above. As a result, for tokens where the model is “confidently wrong” (e.g., “Sydney” in “The capital of Australia is Sydney.”). We argue that although the entropy value might be lower, it remains non-negligible since modern languages do not typically exhibit such extreme sharpness for an informative but wrong token.
>
>    Thus, entropy weighting does not eliminate semantically important gradients; it simply prevents irrelevant tokens (articles, punctuation, filler subwords) from dominating the token-level aggregation.
>
>    We agree that the entropy weight has its own limitations, as suggested by the reviewer (the token 'think' in "I think it’s going to rain" might have a large entropy weight). We choose it as a token weight because of its ***computational efficiency.***
>    - It does not require any additional models or additional computation.
>    - All these values can be obtained along with the responses, which maintain the advantages of the gradient-based method: they can be collected in parallel with the generation process, as stated in line 52.
>
>    While the heuristic is not perfect, our ablation study (Table 2) shows that entropy weighting yields consistent empirical improvements, supporting its practical usefulness.

---

> ### Author Response · Authors · 2025-11-21
> **Author Response III - Response to Weakness of Evaluation**
>
> > ### ***3. Additional Baseline (Weakness of Evaluation)***
>
> Thank you for the suggestion. Below, we report the results of G-NLL. As shown in the table, G-NLL performs reasonably well on datasets with a single ground-truth answer, but its performance drops substantially in high-aleatoric settings.
>
> This behavior is consistent with the intuition behind Parameter Gradient. G-NLL essentially evaluates the negative log-likelihood of the greedy prediction. When the dataset contains only one correct answer, an ideal model concentrates most probability mass on that answer, leading to a low NLL score. However, in high-aleatoric cases, where multiple plausible answers exist for the same input, the correct distribution is inherently multi-modal and does not assign extremely high probability to any single answer. As a result, the greedy prediction’s likelihood no longer reliably reflects correctness, causing G-NLL to become noisy or even misleading as an uncertainty estimate.
>
> We will add G-NLL to our main experiments.
>
> | UQ Methods | Qwen3 & SciQ | Qwen3 & TriviaQ | Qwen3 & TruthfulQ | Mistral & SciQ | Mistral & TriviaQ | Mistral & TruthfulQ | Llama3.1 & SciQ | Llama3.1 & TriviaQ | Llama3.1 & TruthfulQ | Avg. |
> |-----------|-----------------------|---------------------------|-----------------------------|------------------------|---------------------------|-----------------------------|-------------------------------|---------------------------------|-----------------------------------|------|
> | LN-PE     | 67.08 | 80.00 | 64.78 | 76.68 | 84.02 | 66.29 | 72.51 | 84.53 | 63.38 | 73.25 |
> | P(True)   | 57.13 | 76.30 | 49.17 | 71.40 | 81.39 | 53.75 | 64.91 | 78.60 | 54.15 | 65.20 |
> | Self-Con  | 61.95 | 76.64 | 64.26 | 71.07 | 81.80 | 67.03 | 71.47 | 83.56 | 56.78 | 70.51 |
> | Deg       | 65.01 | 78.21 | 63.30 | 74.15 | 83.11 | 67.27 | 73.11 | 84.67 | 59.12 | 71.99 |
> | INSIDE    | 57.96 | 72.47 | 62.29 | 71.54 | 72.56 | 62.21 | 70.83 | 76.24 | 54.50 | 66.73 |
> | S.E.      | 56.88 | 76.16 | 63.10 | 68.53 | 80.64 | 66.71 | 70.27 | 83.12 | 59.59 | 69.45 |
> | S.D.      | 63.79 | 76.41 | 57.60 | 72.52 | 79.07 | 63.11 | 74.00 | 82.44 | 57.75 | 69.63 |
> | M.I.      | 66.25 | 76.26 | 63.75 | 73.72 | 81.88 | 66.06 | 72.43 | 83.52 | 64.25 | 72.01 |
> | SAR       | 72.72 | 81.52 | 67.98 | 76.57 | **85.23** | 68.55 | 75.28 | 85.65 | 64.44 | 75.33 |
> | ExGrad    | 71.34 | 80.37 | 63.77 | **77.53** | 84.53 | 66.40 | 74.11 | 85.22 | 62.00 | 73.92 |
> | G-NLL| 72.70| 81.01| 60.44| 76.83| 84.61| 63.67| 75.49| ***85.91***| 57.51| 73.13|
> | SemGrad   | 72.20 | 80.40 | 69.06 | 75.55 | 82.37 | 72.27 | 75.76 | 84.72 | **69.42** | 75.75 |
> | HybridGrad | **72.83** | **81.69** | **69.61** | 76.90 | 84.13 | **72.72** | **76.31** | 85.89 | 69.25 | **76.59** |
>
> ### ***We hope that our clarifications resolve the reviewer’s concerns. If any aspects of our explanation remain unclear, or if there are further concerns, we would be happy to address them or provide additional experiments. We thank the reviewer again for their time and constructive feedback.***

---

> ### Author Response · Authors · 2025-12-03
> **Summary of Changes in the Revised Submission**
>
> We have revised the original submission based on Reviewer C28m’s valuable feedback and our responses. Specifically:
>
> - We add a paragraph in Section 3.3 (lines 301–309) that explicitly analyzes the strengths and weaknesses of the parameter-gradient method and SemGrad, and explains our motivation for combining them. We also added one sentence in Section 4.3 (lines 402–403) to clarify SemGrad’s weakness in the single-answer setting.
>
> - We add a sentence in Section 3.3 (lines 296–298) explicitly stating that we freeze the entropy weights before computing gradients.
>
> - We incorporated G-NLL into the main experiments, including adding it to the list of baselines (line 349), updating Table 1, and revising Appendix C.2 “Baseline Implementation Details” (lines 870–872).
>
> We sincerely thank Reviewer C28m for their constructive feedback.

---

### Author Response · Authors · 2025-12-03
**Author Summary for Area Chair (Part III)**

> ### Main Concern of Reviewer C28m (score 4)

Reviewer C28m’s review is overall fairly positive: they gave strong subscores of 3 for soundness, 3 for contribution, and 4 for presentation. Their main concern is that “the authors subsequently reintroduce parameter gradients in the HybridGrad variant, presented only briefly at the end of the methodology section. The justification for this addition is unconvincing and lacks theoretical grounding,” as discussed in the Common Concerns section. In our rebuttal and the revised manuscript, we explicitly analyze the respective strengths and weaknesses of SemGrad and parameter-gradient methods and clarify the motivation for HybridGrad in Section 3.3 (lines 301–309).

Another concern is about the token-importance weight based on token entropy introduced in Section 3.3 (see Author Response II (2)). The purpose of this entropy-based weight is to downweight extremely uninformative tokens, such as articles (“the”, “a”), punctuation (",", "."), prepositions (“to”, “by”), and filler subwords (“-ing”). We acknowledge that this heuristic has limitations, but it is computationally efficient and preserves the advantage that gradient signals can be collected in parallel with the generation process. Importantly, our ablation study (Table 2) shows that entropy weighting yields consistent empirical improvements, supporting its practical usefulness despite its simplicity.

We believe that, had the discussion phase continued, this clarification could have led to a more favorable overall assessment of our work.

> ### Main Concern of Reviewer hgm6 (score 4)

Main concerns of Reviewer hgm6 appear to focus on the conceptual clarity of our motivation and core notions. In weakness 1, Reviewer hgm6 suggests a more explicit explanation of what we mean by “semantics”. In weakness 2, they suggest a clearer discussion about the motivation behind SPS. In weakness 4.2, they question our connection between multiple valid answers and high aleatoric uncertainty (e.g., TruthfulQA with multiple correct answers). In our rebuttal, we provide a more explicit clarification of the definitions and roles of “semantics,” “SPS,” and “aleatoric uncertainty” (please refer to Author Response I (1)(2)(3)). We also revised the paper accordingly, as summarized in the “Summary of Changes in the Revised Submission".

Another concern is about the breadth of the evaluation (Weakness 4.1). We agree that a broader experimental suite is generally desirable. Nevertheless, we believe our current experiments are representative and sufficient to validate both the motivation and effectiveness of our method. Reviewer QuTT, for example, describes our setup as “a good selection of baseline methods on three LLMs in the 7B range.” Further discussion of this point can be found in Author Response III (6).

> ### Main Concern of Reviewer QuTT(before-rebuttal score 4; intended post-discussion score 6)

Reviewer QuTT's main concerns include:
 - BEM evaluation metric should be complemented with LLM-as-a-judge
 - Besides AUROC, other evaluation metrics would provide a more complete picture.
 - Motivation from using the parameter gradient norm is understandable, albeit a bit weak.

They also suggested several additional experiments. We addressed these points one by one in our rebuttal, adding further experiments and metrics as requested. Reviewer QuTT acknowledged that “the provided response has alleviated my concerns” and indicated that they would raise their score to 6 to reflect their updated assessment.

***In summary***, we propose the first gradient-based approach to uncertainty quantification for free-form generation in LLMs. Reviewers describe the core motivation and idea as “clear and neat” and “interesting and novel,” and we believe it opens a promising and principled direction: viewing LLM uncertainty through the lens of semantic gradients. Our method is simple, efficient, and empirically effective, and the proposed SPS metric and analysis provide a useful tool for understanding where semantics are preserved in modern LLMs.

In addition, one reviewer explicitly stated they would raise their score to 6 after reading our responses, while the scores of the other two reviewers were assigned before they could consider our rebuttal and clarifications. We therefore respectfully believe that, had the discussion phase continued, our responses and additional analysis would likely have led to a more favorable overall evaluation and that the paper meets the ICLR acceptance bar.

Thank you again for your time and for the careful reassessment of our submission under these challenging circumstances.

Sincerely,
The Authors

---

### Author Response · Authors · 2025-12-03
**Author Summary for Area Chair (Part II)**

> ### Common Concerns
- ***Motivation of reintroducing parameter gradients in HybridGrad*** (C28m: Weakness 1; QuTT:  Weakness 4).
  - ***Sketch of Our Responses*** (Details in Author Response I (1) to Reviewer C28m; Author Response II (4), Author Response VI (1) to Reviewer QuTT):

    While parameter gradients are principally unreliable under high aleatoric cases---where multiple valid responses lead to a multimodal ground-truth distribution---they remain a valid and often competitive measure in single-ground-truth settings. In such low-aleatoric regimes, the ground truth distribution is typically sharp and unimodal, causing the parameter gradient, which measures the sharpness of the output distribution, to align closely with the model’s training objective and yielding greater numerical stability. In contrast, while SemGrad is theoretically well-motivated in both low- and high-aleatoric settings, it operates by identifying hidden states that serve as a proxy for semantic information. These representations are not guaranteed to perfectly isolate all semantic factors, which can introduce additional numerical instability, making it less stable than the parameter gradients in low-entropy cases.

    Therefore, we propose HybridGrad to leverage the theoretical robustness of SemGrad in high aleatoric settings and the numerical stability of parameter gradient in low aleatoric settings. The empirical improvements are consistent with this design. We add a paragraph in Section 3.3 (lines 301–309) that explicitly analyzes the strengths and weaknesses of the parameter-gradient method and SemGrad, and explains our motivation for combining them in the revision.

- ***The motivation of choosing BEM as the correctness metrics and additional experiments with LLM as a Judge*** (hgm6: Weakness 4.2 and Question 2; QuTT:  Weakness 1).
  - ***Sketch of Our Responses*** (Details in Author Response III (5) to Reviewer hgm6; Author Response I (1) to Reviewer QuTT):

     We chose BEM because it is ***reproducible, cost-free, computationally lightweight***, and prior work [1] has shown it to be an effective metric for evaluating the correctness of short-form QA answers. ***We add experiments using LLM-as-a-judge for correctness evaluation to Appendix D.1.***  The rankings and relative trends under BEM and LLM-as-a-judge are highly consistent, and our proposed methods continue to achieve superior performance under both metrics. This suggests that our main conclusions are robust to the choice of correctness metric.

[1] Kamalloo et al. (2023). Evaluating open-domain question answering in the era of large language models. ACL 2023

---

### Author Response · Authors · 2025-12-03
**Author Summary for Area Chair (Part I)**

Dear Area Chair,

We would like to thank you for taking the additional time to assess our submission under the unusual circumstances following the OpenReview leak and the subsequent rollback of reviews and scores. We are aware of the substantial workload this creates and are grateful for your efforts.

Our paper currently has pre-rebuttal scores of 4, 4, 4. During the discussion phase before the rollback, Reviewer QuTT wrote that “the provided response has alleviated my concerns” and stated that they would “***raise my score to 6***” to reflect their updated assessment, while reviewers C28m and hgm6 ***did not*** participate in the discussion phase before the rollback. We respectfully believe that, had the discussion phase continued, our rebuttal and clarifications would likely have led to a more favorable overall evaluation.

> ### Summary of our work:

Our paper proposes ***the first gradient-based approach to uncertainty quantification for free-form generation in LLMs***. Prior work uses parameter gradients, which effectively measure the sharpness of the predictive distribution, to quantify uncertainty in the classification tasks. However, in high-aleatoric regimes (which are common in language), the ground-truth distribution is inherently non-sharp and multi-modal. Even when the learned model closely matches this ground-truth distribution, parameter gradients can remain large, leading to biased uncertainty estimates and potentially misleading. (Details please refer to analysis in Section 2).

To address this, we propose ***SemGrad***, based on a simple assumption about human language: responses should remain stable under semantically equivalent inputs. We translate this stability into the gradient with respect to an embedding that preserves the semantic structure of inputs, i.e., an internal representation that maps semantically equivalent variants of input closer and maps semantically different inputs distant. We refer to these as ***semantic-preserving embeddings***. Our motivation starts from the basic nature of human language and does not depend on assumptions about output sharpness or unimodality, making it more robust in high-aleatoric settings. (Detailed analysis please refer to Section 3.1). Reviewer C28m describes "The motivation for SemGrad is clear and neat", Reviewer hgm6 acknowledges that the concept "is generally appealing and interesting", and Reviewer QuTT called it "an interesting and novel idea."

To identify such embeddings in practice, we propose the ***Semantic Preservation Score (SPS)*** to directly quantify how well a hidden representation preserves semantic structure, by measuring the alignment difference between semantically equivalent paraphrases and semantically different inputs. Our experiments reveal that each model has a consistent token position $t^{ * }$ that best preserves semantics, and the semantic information is mainly preserved in the deeper half of layers. Consequently, SemGrad computes gradients with respect to the hidden states from the top half of layers at $t^{ * }$. (Section 3.2). Reviewer QuTT highlighted this as a “very interesting investigation with SPS, analyzing the specifics of the different used LLMs regarding what layer and token is most useful to extract semantic information.”

While SemGrad is theoretically well-motivated in both low- and high-aleatoric settings, its empirical performance can be less stable than parameter-gradient methods in low-aleatoric regimes, where parameter gradients are also theoretically justified. We therefore propose ***HybridGrad***, which combines SemGrad with the parameter-gradient approach to leverage SemGrad’s robustness in high-aleatoric settings and the numerical stability of parameter gradients in low-aleatoric settings (Details please refer to analysis Section 3.3).

Our experiments span 11 baselines, 3 different LLMs, and 3 datasets with both low- and high-aleatoric characteristics. Our proposed methods achieve the best average performance across baselines. The results show that our proposed methods achieve the best average performance across baselines. Parameter-gradient methods perform well in single-ground-truth settings, where SemGrad achieves comparable results. In high-aleatoric settings, SemGrad maintains strong performance while parameter-gradient methods degrade substantially. The HybridGrad metric delivers consistently strong and stable performance in most settings, achieving the best overall AUROC. Our ablation studies further reveal a strong correlation between the semantic-preserving capability of the hidden states used by SemGrad and its uncertainty-quantification performance, consistent with our motivation (Details please refer to Section 4). Reviewer C28m described the ablations as “insightful,” and Reviewer QuTT noted the “thoughtful ablation studies, rationalizing many of the choices for implementing the method.”

---

### Meta-Review · Area_Chair_esJo · 2025-12-28

**Summary:**

The authors propose SemGrad, a gradient-based UQ method that operates in semantic space to estimate uncertainty for NLG tasks. The method first identifies the highest semantic preserving token of an input (expected highest value of the proposed metric SPS), whose embeddings best represents the entire input’s semantics. Then it computes the gradient of the log-likelihood of the output sequence with respect to the top half layer embeddings of the identified semantic preserving token. This measures how sensitively the model’s output sequence changes when perturbing the input while preserving its semantics. Additionally, the authors also propose HybridGrad, a combination of SemGrad and parameter gradients.

**Reviewer Concerns:**

1, Lack of formal theoretical guarantees for SemGrad / HybridGrad
2, Although the authors softened wording, the core claim that SemGrad operates in “semantic space” is still not formally grounded. The definition of “semantic-preserving perturbations” remains heuristic and model-internal, not theoretically characterized.

3, SPS definition, entropy-based token weighting, choice of layers/tokens, and interpolation scheme all rely on heuristics. Reviewers (hgm6, QuTT) flagged this repeatedly; rebuttal acknowledges but does not fundamentally reduce heuristic reliance.

4, All experiments are short-form QA. Authors explicitly state limitations, but reviewers’ concern that conclusions may not generalize to broader NLG remains unresolved.

**Reviewer Scores:**

remain unchanged

---

### Decision · Program_Chairs · 2026-01-26

Reject